# Stress-Testing Long-Context Language Models with Lifelong ICL and Task Haystack

**Xiaoyue Xu**[*]
Tsinghua University
xiaoyue.xu.me@gmail.com

**Qinyuan Ye**[*]
University of Southern California
qinyuany@usc.edu

**Xiang Ren**
University of Southern California
xiangren@usc.edu

## Abstract

We introduce Lifelong ICL, a problem setting that challenges long-context language models (LMs) to learn a sequence of language tasks through in-context learning (ICL). We further introduce Task Haystack, an evaluation suite dedicated to assessing and diagnosing how long-context LMs utilizes contexts in Lifelong ICL. When given a task instruction and test inputs, long-context LMs are expected to leverage the relevant demonstrations in the Lifelong ICL prompt, avoid distraction and interference from other tasks, and achieve test accuracies that are not significantly worse than those of the Single-task ICL baseline.

Task Haystack draws inspiration from the widely-adopted "needle-in-a-haystack" (NIAH) evaluation, but presents distinct new challenges. It requires models (1) to utilize the contexts at a deeper level, rather than resorting to simple copying and pasting; (2) to navigate through long streams of evolving topics and tasks, proxying the complexities and dynamism of contexts in real-world scenarios. Additionally, Task Haystack inherits the controllability of NIAH, providing model developers with tools and visualizations to identify model vulnerabilities effectively.

We benchmark 14 long-context LMs using Task Haystack, finding that frontier models like GPT-4o still struggle with the setting, failing on 15% of cases on average. Most open-weight models further lack behind by a large margin, with failure rates reaching up to 61%. In our controlled analysis, we identify factors such as distraction and recency bias as contributors to these failure cases. Further, performance declines when task instructions are paraphrased at test time or when ICL demonstrations are repeated excessively, raising concerns about the robustness, instruction understanding, and true context utilization of long-context LMs. We release our code and data to encourage future research that investigates and addresses these limitations.[1]

## 1 Introduction

Recent advances in model architecture [Han et al., 2024, Su et al., 2024], hardware-aware optimization [Dao et al., 2022, Liu et al., 2024b], training procedure [Tworkowski et al., 2023, Liu et al., 2024a], and data engineering [Fu et al., 2024, An et al., 2024] have enabled large language models (LLMs) to handle extended contexts, reaching up to 32 thousand tokens or even millions [Gemini Team, 2024, Anthropic, 2024]. These advancements have opened up new opportunities and potential use

---

[*]Equal Contribution.

[1] INK-USC/Lifelong-ICL  https://inklab.usc.edu/lifelong-icl

38th Conference on Neural Information Processing Systems (NeurIPS 2024) Track on Datasets and Benchmarks.

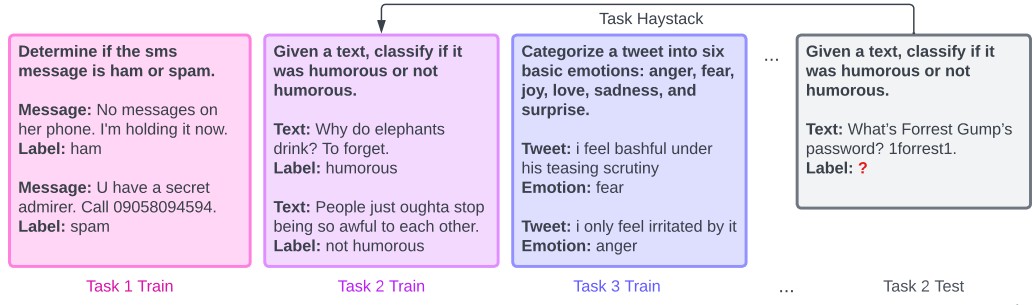

Figure 1: **Lifelong ICL and Task Haystack.** Lifelong ICL presents long-context LMs with a sequence of tasks, each containing a task instruction and a few demonstrations. At test time, the model is given a previously seen task instruction and then makes predictions on the test input directly. A long-context LM "passes" the Task Haystack test when its accuracies in Lifelong ICL (Task 1+2+3) are not significantly worse than accuracies of the Single-task ICL baseline (Task 2 only).

cases for LLMs. However, while long-context LM development strides forward, effective evaluation methods have not kept pace. Systematically evaluating long-context LMs' ability to leverage such long contexts remains an open challenge.

Current evaluation approaches fall into two major categories. The first involves constructing benchmarks with real-world long-context tasks [Shaham et al., 2022, 2023]. While valuable, creating these benchmarks is time-consuming and particularly challenging when scaling the input context length to millions of tokens. The second approach employs synthetic evaluations like the "needle-in-a-haystack" (NIAH) test [Kamradt, 2023] or key-value retrieval tests [Liu et al., 2024c]. For example, in the NIAH evaluation, a piece of information ("The special magic number is 12345") is planted in a haystack of irrelevant contexts (Paul Graham essays; Graham 2024) and the model is evaluated on answering a question about the information ("What's the special magic number?"). Although useful for initial assessment, these tests primarily measure simple copying-and-pasting capabilities and fail to capture whether models are able to utilize the context at a deeper level.

In this work, we offer new perspectives to long-context LM evaluation by introducing Lifelong ICL, a new problem setting that challenges these models to learn a sequence of tasks via in-context learning (ICL). Further, we introduce Task Haystack, an accompanying evaluation suite designed for systematic diagnosis of context utilization (Fig. 1). In Task Haystack, a long-context LM will be evaluated on a collection of tasks, with Lifelong ICL prompts and Single-task ICL prompts respectively. A model "passes" the test if its accuracies with Lifelong ICL prompts are not significantly lower than when using Single-task ICL prompts. The overall pass rate, averaged across tasks and different lifelong stream permutations, serves as the key metric of Task Haystack.

Task Haystack presents unique challenges not fully covered by existing benchmarks. Firstly, Task Haystack requires deeper understanding of the relevant context for accurate predictions. This goes beyond simple retrieval capabilities tested by NIAH-style benchmarks, which often rely on basic copying and pasting. Secondly, Task Haystack features high information density, meaning that every piece of information in the context might be crucial for successful prediction at test time. This differs from evaluation suites in which the important information ("needle") is positioned conspicuously, allowing models to exploit shortcuts [Anthropic, 2024]. Thirdly, existing benchmarks fall short in capturing the dynamics of shifting topics within the context [Zhao et al., 2024], which can pose challenges in real-world applications of long-context models—such as a 24/7 personal assistant that must resume previous conversations amid a long, evolving stream of topics. While not fully realistic, Task Haystack serves as a useful proxy for evaluating this aspect.

We extensively evaluate 14 long-context models on Task Haystack. While all models achieve near-perfect scores on the original NIAH test, none reach satisfactory performance on our proposed evaluation. Among the compared models, GPT-4o and Gemini-1.5-Flash lead with an average pass rate of 85%, significantly outperforming most open-weight models. Llama-3.1-70B, the best-performing open-weight model, follows closely with an average pass rate of 80%. To understand the root causes behind these failure cases, we conduct controlled experiments that isolate factors like recency bias (models favoring information at the end of the context) and distractability (models

getting distracted by irrelevant information). The results confirm that both factors contribute to performance degradation on Task Haystack. Additionally, we find that model performance declines when instructions are paraphrased at test time and when few-shot ICL demonstrations of a single task are repeated multiple times. These observations highlight the limitations of current long-context LMs in terms of robustness, instruction understanding, and context utilization.

We hope that Lifelong ICL and Task Haystack serve as useful resources and testbeds for evaluating, diagnosing, and understanding long-context LMs. Further, we anticipate that the limitations and vulnerabilities exposed in this paper will inspire innovations in long-context LM development.

## 2 Related Work

**Long-Context LM Evaluation.** Early studies on long-context modeling primarily rely on perplexity-based evaluations [Beltagy et al., 2020, Press et al., 2022]. Subsequent research has indicated that such evaluation is limited in reflecting a model's effectiveness in downstream applications [Sun et al., 2021, Hu et al., 2024]. Recent efforts have led to the development of comprehensive benchmarks for evaluating long-context models, which can be divided into realistic and synthetic categories. Realistic benchmarks, exemplified by (Zero)SCROLLS [Shaham et al., 2022, 2023], comprise tasks that require processing long inputs collected from real-world scenarios. These tasks are typically sourced from established datasets and include various task types such as summarization and question answering, or developed from inherently lengthy corpus such as novel [Zhang et al., 2024], grammar books [Tanzer et al., 2024] and code repository [Jimenez et al., 2024]. In the category of synthetic benchmarks, the needle-in-a-haystack (NIAH) [Kamradt, 2023] evaluation is widely adopted for evaluating context utilization [Gemini Team, 2024, Anthropic, 2024, Liu et al., 2024a, Fu et al., 2024, Levy et al., 2024, *i.a.*]. Ruler [Hsieh et al., 2024] expands on the NIAH test with multi-key and multi-value retrieval, and adds two new tasks that involve multi-hop tracing and aggregation. Hybrid benchmarks is a middle-field that incorporate both realistic and synthetic elements. An example is LongBench [Bai et al., 2024], which includes synthetic tasks based on realistic text, such as counting unique passages appearing in the context. Our proposed Task Haystack can be seen as a hybrid benchmark, with a realistic touch as (1) it is built upon realistic language tasks; (2) it proximates the challenge of navigating through evolving topics and tasks.

**Evaluating Long-Context LMs with Many-Shot ICL.** Several recent works have explored in-context learning with long-context LMs by scaling the number of training examples (*i.e.*, shots). Bertsch et al. [2024] conducted a systematic study of long-context ICL with up to 2,000 shots, demonstrating many-shot ICL as a competitive alternative to retrieval-based ICL and fine-tuning. Additionally, it offers the advantage of caching demonstrations at inference time, unlike instance-level retrieval methods. While Bertsch et al. [2024] focus on classification tasks, Agarwal et al. [2024] showed the effectiveness of many-shot ICL on generative and reasoning tasks, and established new state-of-the-art results on practical applications such as low-resource translation with the Gemini 1.5 Pro model. However, there are still limitations to many-shot ICL. Li et al. [2024] introduce LongICLBench, a suite of 6 classification tasks with many (20+) classes, and find that current long-context LMs still struggle with these tasks. Orthogonal to this line of work on scaling *number of examples* for one single task, we focus on scaling the *number of tasks* in our Lifelong ICL setting.

**Lifelong Learning in NLP.** Lifelong learning, or continual learning, refers to the problem setting where a model learns continuously from data streams [Biesialska et al., 2020, Shi et al., 2024]. Lifelong ICL is largely inspired by this line of work and challenges long-context models to learn continuously from a sequence of language tasks. However, unlike prior works that use gradient-based fine-tuning [de Masson d'Autume et al., 2019, Jin et al., 2021, Scialom et al., 2022, Mehta et al., 2023], Lifelong ICL is a new exploration that uses in-context learning as the underlying "learning" algorithm. It also stands out from Coda-Forno et al. [2023] and Ye et al. [2024] by focusing on evaluating long-context LMs and scaling the input length from 4k to up to 32k tokens. A primary challenge in lifelong learning is catastrophic forgetting, the tendency of a model to forget previously acquired tasks upon learning new tasks [Kirkpatrick et al., 2017]. Our proposed Task Haystack evaluation focuses an analogous phenomenon, as the model may struggle to recall earlier information in a lengthy context, leading to a performance decline.

# 3    Problem Setting

In the following, we establish the notations and the problem setting of Lifelong ICL in §3.1. We will begin by defining notations of in-context learning (ICL) of *one single task $T$*. We will then build upon these foundations and introduce Lifelong ICL with *a collection of tasks $\mathcal{T}$*. In §3.2, we further introduce our Task Haystack evaluation protocol, provide the definition of the key metric named "pass rate," and describe our strategies to account for the instabilities in ICL experiments.

## 3.1    Lifelong ICL

**In-context Learning.**    In-context learning is a method that adapts LMs to perform a language task by providing prompts containing input-output pairs [Brown et al., 2020]. In this paper, we define a language task $T$ as a tuple of $(D^{train}, D^{test}, d)$, where $D^{train}$ is the training set, $D^{test}$ is the test set, $d$ is a textual task description (*i.e.*, instruction). We first create a task-specific prompt $p$ by concatenating the task description and the $k$-shot examples in $D^{train}$, *i.e.*, $p = d \oplus x_1^{train} \oplus y_1^{train} \oplus \ldots \oplus x_k^{train} \oplus y_k^{train}$. Then, to make a prediction on the test input $x^{test}$, we concatenate the task-specific prompt and the test input (*i.e.*, $p \oplus x^{test}$), and query the language model LM to generate the prediction $\hat{y}$. We denote this process as $\hat{y} = \text{LM}(x^{test}|p)$ to highlight that the prediction is made by conditioning on the task-specific prompt $p$.

**Task Collection and Task Permutation.**    The definition above introduces how ICL is performed with one single task $T$. In Lifelong ICL, an LM is expected to learn from a collection of $n$ tasks, denoted as $\mathcal{T} = \{T_i\}_{i=1}^n$. To enable this, we first create a random permutation $a = (a_1, a_2, \ldots, a_n)$, thus the tasks in $\mathcal{T}$ will be ordered as $(T_{a_1}, T_{a_2}, \ldots, T_{a_n})$. For example, when $n = 3$, one possible permutation $a$ is $(3, 1, 2)$, so that the tasks are ordered as $(T_3, T_1, T_2)$.

**Lifelong ICL.**    Given a permutation $a$, we first create the task-specific prompt $p_{a_i}$ for each task $T_{a_i}$, and then create the Lifelong ICL prompt $p_l$ by concatenating all task-specific prompts, *i.e.*, $p_l = p_{a_1} \oplus p_{a_2} \oplus \ldots \oplus p_{a_n}$. At test time, for *each* task $T_{a_i}$ in $\mathcal{T}$, the model will be queried to perform generate the prediction as $\hat{y} = \text{LM}(x_{test}|p_l \oplus d_{a_i})$. Note that we append the task description $d_{a_i}$ after the Lifelong ICL prompt $p_l$ at test time, to ensure the model is informed of the task at hand. See Fig. 1 for an illustrative example with 3 tasks.

## 3.2    Task Haystack

**Evaluation Principle.**    For a test task $T_{a_i}$, we anticipate that long-context LMs can effectively utilize the in-context examples of that task, *i.e.*, $p_{a_i}$, which is a substring of the Lifelong ICL prompt $p_l \oplus d_{a_i}$. To evaluate this, we compare the model performance on task $T_{a_i}$ when conditioning on $p_l \oplus d_{a_i}$ and $p_{a_i}$, and expect the former to be not significantly worse than the latter. In other words, the Single-task ICL prompt $p_{a_i}$ is the "needle" in the Lifelong ICL prompt $p_l$ (*i.e.*, the "task haystack").[2]

**Addressing ICL Instability with Multiple Runs.**    One challenge in Task Haystack evaluation is the notorious instability of ICL. To account for this, our experiments will be carried out with 5 random samples of the permutation $a$ and 5 randomly-sampled few-shot training set $D_{train}$ for each task. This allows us to obtain a performance matrix of size $(t, p, r)$ for Lifelong ICL, where $t$ is the task index, $p$ is the permutation index, and $r$ is the few-shot sample index.[3] We will also obtain a matrix of size $(t, r)$ for the Single-task ICL baseline.

**Evaluation Metrics.**    For an **overall measurement**, we introduce an **overall pass rate**. For each permutation $a$ and each task $T_{a_i}$, we will get two groups of performance metrics, when using Single-task ICL and Lifelong ICL respectively. Each group contains 5 metrics, corresponding to the 5 randomly-sampled few-shot training set $D_{train}$. The model passes the test (*i.e.*, scores 1) when the

---

[2]While the *absolute* performance in Lifelong ICL will be influenced by various factors, such as the LM's core capabilities, its parametric knowledge, the prompt template, or the selection of ICL examples, it is reasonable to make a *comparative* assumption that the performance of Lifelong ICL should not be worse than Single-task ICL for the same model. Additionally, since the Single-task ICL prompt $p_{a_i}$ is a substring of the Lifelong ICL $p_l$, the quality of the ICL examples are controlled to be the same.

[3]In a Task Haystack of 16 tasks, we run 16*5*5=400 experiments and obtain 400 performance metrics.

the Lifelong ICL group is not significantly worse than the Single-task ICL group, captured by a two-sided t-test with $p = 0.05$. The model scores 0 otherwise. The overall pass rate will be computed by averaging the scores over the different permutations and tasks. We provide more details of the definition and discuss its limitations in §A.3.

For a **fine-grained analysis**, our experiment results allow us to **visualize the pass rates grouped by the position in the task stream, by the task, or by the task permutation**. This enables straightforward visualizations as popularized by the needle-in-a-haystack test, providing an convenient tool to diagnose and uncover the vulnerabilities of long-context LMs. See Fig. 24 for an example.

# 4   Experiment Details

**Task Selection.**   While the problem setting in §3 is generic and admits any language task, in this work we instantiate the setting with a narrower task distribution for initial exploration. Our key considerations include:[4]

- We focus on classification tasks, as they allow standardized evaluation. Additionally, a large body of past work investigates ICL empirically or mechanistically using classification tasks [Halawi et al., 2023, Chang and Jia, 2023, Wang et al., 2023, Chang et al., 2024, *i.a.*].
- We select classification tasks with fewer than 20 categories and input text shorter than 1000 tokens, to avoid excessively long single-task prompts that dominate the whole context window [Li et al., 2024].
- We focus on English tasks, since most long-context LMs are not optimized for multilingual usage.

After careful manual selection, we obtain a collection of 64 classification tasks, covering a wide range of domains and label spaces. We provide a snippet of 16 tasks in Table 1 and provide detailed descriptions of all 64 tasks, including their references and license information, in Table 6.

Table 1: **A Snippet of 16 tasks used in our experiments.** See Table 6 for the full list of 64 tasks. The 16 tasks in this table are used for the Scale-Shot experiments in Table 2.

| | | | |
|---|---|---|---|
| emo | covid_fake_news | logical_fallacy_detection | dbpedia_14 |
| amazon_massive_scenario | news_data | semeval_absa_restaurant | amazon_counterfactual_en |
| brag_action | boolq | this_is_not_a_dataset | insincere_questions |
| clickbait | yahoo_answers_topics | pun_detection | wiki_qa |

**Models.**   We evaluate eleven open-weight long-context LMs on Task Haystack: Mistral-7B (32k) [Jiang et al., 2023], FILM-7B (32k) [An et al., 2024], Llama-2-7B (32k) [TogetherAI, 2024], Llama-2-7B (80k) [Fu et al., 2024], Llama-3-8B/70B (1048k) [GradientAI, 2024a,b], Llama-3.1-70B (128k) [Dubey et al., 2024], Yi-6B/9B/34B (200k) [01.AI et al., 2024], and Command-R-35B (128k) [Cohere for AI, 2024]. These models represent various long-context modeling techniques, model size, and base pre-trained models. Additionally, we evaluate three closed models, GPT-3.5-Turbo (16k) and GPT-4o (128k) from OpenAI, and Gemini-1.5-Flash (1048k) from Google DeepMind [Gemini Team, 2024]. We provide the detailed descriptions of these models in Table 5 in §A.1.

**Controlling the Context Length.**   We consider creating long contexts with two strategies: **(1) Scale-Shot**: scaling the number of in-context examples ($n_{shot}$); **(2) Scale-Task**: scaling the number of tasks ($n_{task}$). In the first setting, we fix $n_{task} = 16$ and experiment with $n_{shot} \in \{1, 2, 3, 4, 5, 6, 7, 8\}$. We use the 16 tasks listed in Table 1 in the main body of the paper.[5]  In the second setting, we fix $n_{shot} = 2$ and experiment with $n_{task} \in \{8, 16, 24, 32, 40, 48, 56, 64\}$. Note that to ensure the in-context examples are balanced and every class is covered, $n_{shot} = 2$ refers to using 2 examples *per class* for in-context learning. In both scaling settings, we are able to effectively create contexts of sizes ranging from 4k to 32k tokens.[6]

We defer additional implementation and engineering details in §A.4.

---

[4]We discuss the limitations of these design choices in §6. We invite future work to improve upon our work and address these limitations.

[5]Following reviewer feedback, we create a separate subset of 16 tasks with permissive licenses, and report the results on selected models in §B.2. We recommend using this subset for future benchmarking and analysis.

[6]It is possible to further increase the context length, *e.g.*, reaching 128k tokens with 64 tasks and 8 shots. Due to compute constraints, we limit the context length to 32k in this work.

Table 2: **Main Results: Fixing 16 Tasks, Scaling the Number of Shots**. "S-acc" stands for Single-task ICL accuracy averaged over all 16 tasks, and "L-acc" stands for Lifelong ICL accuracy. "pass" represents the "pass rate" defined in §3.2, *i.e.*, percentage of cases that Lifelong ICL is not significantly worse than Single-task ICL among 5 random samples of few-shot training sets. L-acc is expected to be not worse than S-acc, and the pass rate is expected to be close to 100%.

| Model | 0-shot | 1-shot (4k) | | | 2-shot (8k) | | | 4-shot (16k) | | | 8-shot (32k) | | |
|---|---|---|---|---|---|---|---|---|---|---|---|---|---|
| | S-acc | S-acc | L-acc | pass | S-acc | L-acc | pass | S-acc | L-acc | pass | S-acc | L-acc | pass |
| Mistral-7B (32k) | 68.1 | 73.9 | 74.6 | 91.2 | 77.6 | 74.6 | 73.8 | 78.6 | 74.8 | 67.5 | 80.3 | 74.2 | 47.5 |
| FILM-7B (32k) | 71.1 | 76.7 | 74.7 | 77.5 | 79.1 | 75.1 | 77.5 | 79.6 | 75.4 | 72.5 | 80.8 | 74.9 | 55.0 |
| Llama-2-7B (32k) | 61.9 | 69.8 | 63.3 | 77.5 | 72.8 | 64.5 | 53.8 | 75.6 | 63.0 | 41.2 | 78.0 | - | - |
| Llama-2-7B (80k) | 38.4 | 47.6 | 60.0 | 100.0 | 49.8 | 60.2 | 100.0 | 56.3 | 62.3 | 96.3 | 59.8 | 61.5 | 76.3 |
| Llama-3-8B (1048k) | 51.2 | 65.5 | 68.1 | 78.8 | 70.0 | 69.1 | 76.2 | 71.5 | 70.1 | 71.3 | 73.6 | 70.1 | 57.5 |
| Llama-3-70B (1048k) | 60.7 | 79.1 | 72.9 | 68.8 | 79.0 | 74.4 | 50.0 | 80.3 | 75.3 | 57.5 | 81.7 | 75.7 | 51.2 |
| Llama-3.1-70B (128k) | 58.8 | 81.7 | 81.2 | 80.0 | 82.8 | 81.1 | 76.2 | 84.6 | 82.4 | 83.8 | 85.2 | 83.3 | 80.0 |
| Cmd-R-35B (128k) | 65.6 | 73.0 | 74.6 | 81.2 | 75.3 | 75.5 | 61.3 | 78.9 | 75.6 | 52.5 | 80.5 | 75.3 | 41.2 |
| Yi-6B (200k) | 51.3 | 70.1 | 57.9 | 61.3 | 73.0 | 58.6 | 51.2 | 75.0 | 58.4 | 43.8 | 75.5 | 57.7 | 38.8 |
| Yi-9B (200k) | 57.0 | 74.5 | 71.5 | 71.2 | 77.7 | 72.9 | 71.2 | 78.0 | 72.9 | 63.7 | 80.0 | 72.9 | 47.5 |
| Yi-34B (200k) | 63.1 | 74.1 | 71.7 | 62.5 | 74.1 | 72.4 | 60.0 | 76.1 | 72.9 | 63.8 | 78.2 | 72.6 | 53.8 |
| GPT-3.5-Turbo (16k) | 78.3 | 81.6 | 76.3 | 73.8 | 82.6 | 79.6 | 71.3 | 83.2 | 79.5 | 62.5 | 81.8 | - | - |
| GPT-4o (128k) | 70.7 | 85.8 | 87.4 | 86.3 | 87.0 | 87.8 | 81.3 | 87.0 | 88.4 | 83.8 | 87.5 | 89.1 | 88.8 |
| Gemini-1.5-Flash (1048k) | 63.7 | 78.0 | 79.1 | 87.5 | 77.9 | 79.4 | 87.5 | 79.4 | 80.4 | 85.0 | 77.9 | 81.6 | 80.0 |

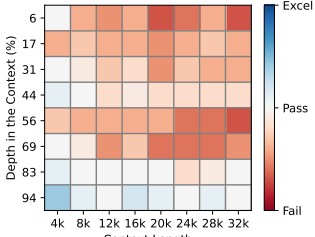

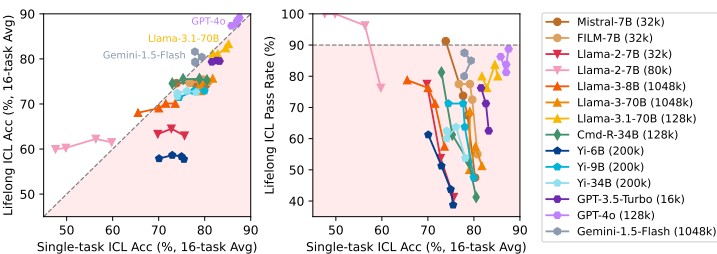

Figure 2: **Task Haystack Results with FILM-7B (32k)** (N-task=16, N-shot=1,2,...,8) visualized in the needle-in-a-haystack style heatmap.

Figure 3: **Visualizing Lifelong ICL accuracy (L-acc) and pass rate as a function of single-task ICL accuracy (S-acc).** Each line is constructed by varying the number of shots in {1,2,4,8} while fixing 16 tasks. Most models fall into the undesired (light red) area. GPT-4o shows the strongest overall performance in our evaluation.

## 5 Results and Analysis

### 5.1 Main Results

**Long-context LMs struggle in Task Haystack.** We present the aggregated results (mean accuracy and overall pass rate) of the Scale-Shot setting in Table 2 and the results of the Scale-Task setting in Table 7. The overall pass rates fall below 90% in 50 out of 54 cases reported in Table 2 and in 41 out of 44 cases in Table 7. When scaling to 32k context with 8 shots and 16 tasks, 9 out of the 11 open-weight models achieve pass rates lower than 60%, suggesting that these models are still far from fully utilizing and flexibly conditioning on the provided context. In the most extreme case, Yi-6B (200k) achieves a pass rate of merely 38.8% in the 8-shot (32k) setting.

While model developers commonly use near-perfect needle-in-a-haystack results as evidence of successful long context utilization [01.AI et al., 2024, GradientAI, 2024b,a], our Task Haystack exposes previously unknown limitations of these models and suggest that these models are far from perfect when deeper, contextual understanding is required.

**A Holistic View of Accuracies and Pass Rates.** One advantage of the pass rate metric introduced in §3.2 is that it isolates the long-context modeling capabilities from models' core capabilities. However, using pass rate as the only metric may inadvertently create a shortcut where a model can achieve perfect pass rates by simply performing poorly in both Single-task ICL and Lifelong ICL.

To have a holistic view on this, we visualize the results from our Scale-Shot experiments by plotting the Lifelong ICL accuracy and pass rate as a function of Single-task ICL accuracy in Fig. 3. For

Table 3: **Summary of Controlled Settings.** "T1 Train" contains the task instruction and few-shot demonstrations of Task 1. "T1 Test" contains the same task instruction and one test input. For the Random setting, we use Paul Graham essays [Graham, 2024] as the random text. In Random and Repeat settings, the input context lengths are controlled to be comparable with the Recall setting. ⤨ = shuffling the few-shot examples; ↻ = using a paraphrased instruction $d'$ at test time.

| Setting | Input Prompt Example | Controlled Factors | | |
|---|---|---|---|---|
| | | Long Ctx. | Distraction | Recency |
| **Baseline (Single-task ICL)** | T1 Train T1 Test | ✗ | ✗ | ✓ |
| Random | Random Text T1 Train T1 Test | ✓ | ✓ | ✓ |
| Repeat | T1 Train T1 Train T1 Train T1 Test | ✓ | ✗ | ✓ |
| Repeat+Shuffle | T1 Train ⤨ T1 Train ⤨ T1 Train T1 Test | ✓ | ✗ | ✓ |
| **Recall (Lifelong ICL)** | T1 Train T2 Train T3 Train T1 Test | ✓ | ✓ | ✗ |
| Replay | T1 Train T2 Train T3 Train T1 Train T1 Test | ✓ | ✓ | ✓ |
| Remove | T2 Train T3 Train T1 Test | ✓ | ✓ | N/A |
| Paraphrase | T1 Train T2 Train T3 Train ↻ T1 Test | ✓ | ✓ | ✗ |

nearly all models, pass rates decrease when the context length increases, highlighting that while these models are able to take in long context as inputs, they are not necessarily utilizing them effectively. For model-wise comparison, GPT-4o takes the lead in terms of both the ICL accuracy and the pass rate. Llama-3.1-70B stands out as the leading open-weight model, achieving an average pass rate of 80%, which is close to the 85% pass rates of GPT-4o and Gemini-1.5-Flash.

One outlier that we notice is the Llama-2-7B (80k) model [Fu et al., 2024], which achieves low ICL accuracies but high pass rates. We notice that this model is trained on language modeling objectives without further instruction tuning or RLHF, which may be the reason behind this trend. This observation also suggests that the pass rates should always be considered together with metrics representing the model's core capabilities.

**Visualization and Diagnostic Tool for Task Haystack.** Task Haystack supports straightforward visualization for diagnosing model vulnerabilities. In Fig. 2 we present the results of Task Haystack (Scale-Shot Setting) in a way similar to the original needle-in-a-haystack (NIAH) evaluation. While FILM-7B achieves near-perfect results in the original NIAH eval, Fig. 2 suggests that it's vulnerable when the context length exceeds 12k, particularly when the relevant information appears in the first 75% of the context window. We include NIAH-style visualizations for all compared models in Fig. 9-22. In addition, we provide examples of aggregating results by permutations, by depth in the context, and by task in Fig. 25-29. We further discuss our findings in §E.2.

### 5.2 Controlled Analysis on Long-Context Utilization

Previously, we find that long-context LMs struggle in the Task Haystack evaluation. In the following section, we investigate the reasons that contribute to their failures with various controlled analyses.

We hypothesize that the model failure at Lifelong ICL may be associated with the following factors: **(a) Recency Bias**: the model mainly relies on recent context and performs worse when the relevant context is distant; **(b) Distraction**: the model may be confused by irrelevant context; **(c) Long-context Inputs**: the model tend to break in general when the input text is long.

Based on these hypotheses, we introduce controlled settings, such as *replaying* the test task at the end of the Lifelong ICL prompt, or *repeating* the Single-task ICL prompt multiple times. Additionally, we use *paraphrases* of the instructions to investigate the model's sensitivity. We summarize these controlled settings in Table 3 and conduct experiments in the 16-task 4-shot setting with Mistral-7B (32k) and FILM-7B (32k). We present the results in Fig. 4 and discuss our findings below.

**(a) Recency Bias.** We investigate the effect of recency bias by comparing the results of Recall and Replay. By replaying ICL demonstrations immediately before testing, model's accuracy improves by 1.6% for Mistral-7B and 2.9% for FILM-7B. Replay can be also considered as an oracle for potential mitigating strategies such as prompting the model to recall relevant information [Shi et al., 2023,

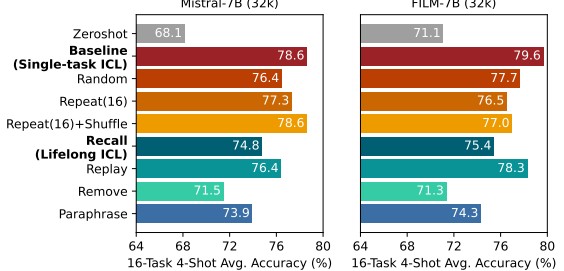
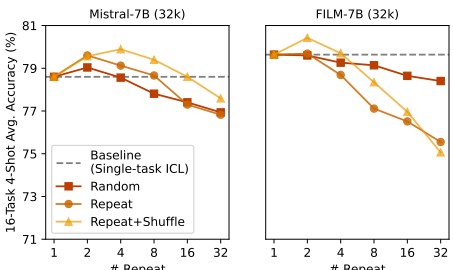

Figure 4: **Controlled Experiments.** Results suggest that long-context LMs are subject to various robustness problems. See §5.2 for discussion.

Figure 5: **Single-Task "Multi-epoch" ICL.** Model performance improves then degrades after repeating ICL examples.

Anthropic, 2024]. However, the improvements only close about half the gap between Baseline and Recall, suggesting that recency bias contributes to but does not fully explain the performance gap.

**(b) Distraction.** We examine the effect of irrelevant context, by contrasting Baseline with Random. The results indicate that prepending an irrelevant long text will influence the performance negatively, which corroborates with recent work investigating the robustness of language models [Levy et al., 2024]. Further, Replay can be seen as prepending a long prefix of mostly irrelevant tasks before performing Single-task ICL (Baseline), and thus the gap between Replay and Baseline may be interpreted as caused by prepending irrelevant contexts.

**(c) Long-context Input.** We further compare Baseline, Random, Repeat settings altogether, where Random introduces *irrelevant* context and Repeat includes only *relevant* context. Perhaps surprisingly, performance drops in the Repeat setting (-1.3% for Mistral-7B and -3.1% for FILM-7B), where both distractions and recency biases are absent. This observation raises concerns on whether longer inputs are more likely to trigger failure modes and give rise to undesired behaviors in general. While more evidence is needed to derive a conclusion, we suggest that long-context LM users be cautious about including everything in the context window, and we recommend using external filtering or retrieval models when necessary.

**Dependency on Task Instructions and ICL Demonstrations.** In the Remove setting, we remove the task instruction and the ICL examples of the test task from the Lifelong ICL prompt, to investigate whether the models are relying on such information. We observe a clear performance drop in the Remove setting (-3.3% for Mistral-7B and -4.1% for FILM-7B compared to Recall), suggesting that the models are able to locate and make use of the "needle" to some extent in the Recall setting, but not doing it precisely so that the performance can match with the Single-task ICL baseline.

The Paraphrase setting further allows us to explore how models make use of task instructions. We observe a decline in performance in the Paraphrase setting compared to Recall. This confirms that the models locate the "needle" by retrieving identical instructions in the context. However, the performance gap indicates that models mainly rely on pattern matching rather than deeper understanding of the instructions, which might limit their broader utility in practical applications.

**Repeated ICL as "Multi-epoch" ICL.** We conduct further investigation with the Random, Repeat, Repeat+Shuffle setting, by varying the size of the context and the number of repetitions. Results are reported in Fig. 5. Interestingly, model performance first increases and then dips when running in-context learning for multiple "epochs." One direct takeaway is that repeating the ICL examples multiple times can potentially improve performance, which may have practical utilities in certain low-data high-inference-budget regimes. However, model performance starts to degrade after repeating more than 8 times. This phenomenon can be interpreted in two ways: (1) It is a known issue that repetition may lead to model degeneration [Nasr et al., 2023]; Repeat+Shuffle can possibly alleviate this issue by introducing slight variations in each repeat, which explains why Repeat+Shuffle outperforms Repeat in general. (2) It is also possible that the model "overfits" to the few-shot training data after multiple "epochs", analogous to the common observations in gradient-based fine-tuning. We invite future work to investigate the working mechanism of ICL in this "multi-epoch" setting.

## 5.3 Additional Observations and Analysis

**Tasked learned via ICL are more easily influenced.** While examining Task Haystack results, we find that the passing and failing behaviors are highly task-specific. For example, in Fig. 24, Mistral-7B (32k) fails on `news_data` and `insincere_questions` in all permutations, meanwhile passes on more popular tasks like `boolq` and `yahoo_answer_topics`. We hypothesize that models may have memorized some of the tasks during pre-training or post-training, making these tasks less subjective to performance drop in Lifelong ICL. Alternatively, a task may be too challenging for the model to learn through ICL, and thus it passes the test by maintaining low performance in both Single-task ICL and Lifelong ICL settings.

To account for these situations, we split all tasks into 2 groups for each model. Tasks of which 4-shot performance is significantly better than 1-shot performance are classified as ICL-effective tasks, and the remaining tasks are considered to be ICL-ineffective. We report the pass rates for each model on these two groups in Table 4. For 10 out of 12 models, pass rates on ICL-effective tasks are lower than pass rates on ICL-ineffective tasks, suggesting that these models tend to "forget" tasks that are newly acquired, and that the overall pass rates may be an overestimate.

Table 4: **Pass Rates on ICL-effective/ineffective Tasks.** Results are computed in the 16-task 4-shot setting. We define ICL-effective tasks as tasks whose 4-shot performance is significantly better than its 1-shot performance. In general, ICL-effective tasks have lower pass rates.

| Model | ICL-eff. N | ICL-eff. pass | ICL-ineff. N | ICL-ineff. pass | All pass | Model | ICL-eff. N | ICL-eff. pass | ICL-ineff. N | ICL-ineff. pass | All pass |
|---|---|---|---|---|---|---|---|---|---|---|---|
| Mistral-7B (32k) | 5 | 36.0 | 11 | 81.8 | 67.5 | Cmd-R-35B (128k) | 5 | 40.0 | 11 | 58.2 | 52.5 |
| FILM-7B (32k) | 2 | 40.0 | 14 | 77.1 | 72.5 | Yi-6B (200k) | 6 | 46.6 | 10 | 42.0 | 43.8 |
| Llama-2-7B (32k) | 6 | 33.3 | 10 | 46.0 | 41.2 | Yi-9B (200k) | 6 | 50.0 | 10 | 72.0 | 63.7 |
| Llama-2-7B (80k) | 3 | 80.0 | 13 | 100.0 | 96.3 | Yi-34B (200k) | 3 | 46.7 | 13 | 67.7 | 63.8 |
| Llama-3-8B (1048k) | 6 | 40.0 | 10 | 90.0 | 71.3 | GPT-3.5-Turbo (16k) | 5 | 44.0 | 11 | 70.9 | 62.5 |
| Llama-3-70B (1048k) | 4 | 35.0 | 12 | 65.0 | 57.5 | GPT-4o (128k) | 6 | 96.7 | 10 | 76.0 | 83.7 |

**Trends of positive task transfer.** While our study mainly focus on undesired performance degradation in Lifelong ICL, which is analogous to the catastrophic forgetting phenomenon in lifelong learning, we also observe trends of positive forward and backward transfers, two desired properties of lifelong learning.[7] In our pass rate design, we deliberately choose *two-sided* t-test to account for both performance gains and drops. We observe positive transfers in Fig. 2, represented by the blue-colored cells in the 1-shot (4k) column and the last row (94% depth). Similar observations can be made with Llama-2 (32k) in Fig. 12 and GPT-4o in Fig. 22. Additionally, Mistral-7B achieves +3.4% performance gain in the Remove setting compared to the Zero-shot baseline (Fig. 4). We consider these as initial evidence for positive transfer in Lifelong ICL, and invite more rigorous analysis to further explore the properties of Lifelong ICL.

## 6 Discussion

**Intended Use.** We anticipate Lifelong ICL and Task Haystack to be used for evaluating and diagnosing newly released long-context LMs. However, as our findings in Sections 5.1 and 5.3 suggest, the ICL accuracy and pass rate might be affected if the model has been trained on the tasks used in our evaluation. To ensure responsible use, we encourage users to (1) investigate and report any potential data contamination; (2) report pass rates on ICL-effective/ineffective groups respectively, as done in §5.3. Additionally, it is possible to use targeted data engineering to improve pass rates on Task Haystack. For fair comparisons, we recommend that users disclose whether their training data contains sequences in a format similar to Task Haystack evaluation.

**Limitations.** (1) As an initial exploration in the Lifelong ICL setting, we primarily focuses on English-only text classification tasks. This potentially limits a comprehensive assessment of model capabilities across various challenges. To get a more complete picture, the evaluation suite may be

---

[7]Forward transfer occurs when "a model reuses knowledge acquired from previous tasks when learning new ones"; backward transfer refers the the phenomenon that "a model achieves improved performance on previous tasks after learning a new task." [Biesialska et al., 2020]

improved by including more diverse tasks categories (*e.g.*, question answering, conditional generation [Ye et al., 2021]), modalities (*e.g.*, vision [Sharma et al., 2024], speech), and languages. We encourage future research to build upon our foundation and explore these more complex settings. (2) This work simplifies the lifelong learning stream by assuming a sequential order, clear task boundaries, and a fixed number of examples per class for each task. Real-world scenarios likely involve a more dynamic learning stream, without clear task boundaries or assumptions on the sequential order. In §B.6, we conduct preliminary experiments by interleaving examples of multiple tasks in the context. Future work may explore more realistic lifelong learning streams with increased complexity. (3) Finally, due to computational constraints, our evaluation utilizes 5 random permutations of tasks order and 5 different random samples of few-shot training sets. Experimenting with a larger number of samples could potentially reduce the randomness inherent in the results and increase the reliability of the findings. Additionally, we limit our evaluation to up to 32k input tokens. Stress-testing long-context models with their full context lengths may reveal further limitations of these models.

**Ethics Statement.** This work leverages openly available datasets that were carefully reviewed by the authors to mitigate potential data privacy and security concerns. To the best of our knowledge, the datasets we use do not contain personally identifiable information. Some datasets contain offensive content when the underlying task is offensive content (*e.g.*, hate speech) classification. We emphasize that these datasets are used solely for evaluation purposes. As our research does not involve model training or the release of new models, the risk of amplifying biases within the data is minimal.

# 7 Conclusion

In this paper, we introduced Lifelong ICL, a novel problem setting for long-context LMs, and developed Task Haystack, a concrete evaluation suite focusing on evaluating and diagnosing long-context LMs in the Lifelong ICL setting. Our experiments with 14 long-context LMs revealed that while these models excel at needle-in-a-haystack style evaluation, their ability to utilize the context flexibly and contextually remains limited. Through our controlled analysis, we dissected and quantified factors such as recency biases and distractions that contribute to performance drops. We also identified performance degradation when repeating ICL examples or using paraphrased instructions, highlighting a fundamental vulnerability in current long-context models.

Our results demonstrate that Task Haystack still poses significant challenges for newly-released long-context models. We hope that Lifelong ICL and Task Haystack will serve as valuable tools for diagnosing and advancing the development of future long-context LMs. Additionally, we consider our work as an exploratory step towards backprop-free algorithms in lifelong learning settings.

# Acknowledgment

We thank the anonymous reviewers for their thoughtful feedback and active engagement throughout the discussion period. In addition, we thank Xisen Jin, Jun Yan, Ting-Yun Chang, Daniel Firebanks-Quevedo, Johnny Wei, Ryan Wang, Wenbo Zhang for insightful discussions. This work was supported in part by Cohere For AI Research Grant Program and OpenAI Researcher Access Program. Qinyuan Ye was supported by a USC Annenberg Fellowship.

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

# A  Experiment Details

## A.1  Models

We list the details of open-weighted models evaluated in Table 5. For closed models from OpenAI, the specific model versions we evaluated are `gpt-3.5-turbo-0125` and `gpt-4o-2024-05-13`. For Gemini 1.5 Flash, we evaluated `gemini-1.5-flash-001`.

Table 5: **Open-weight Long-context LMs Evaluated in This Work.**

| Model | Max $L$ | Reference | Huggingface Identifier |
|---|---|---|---|
| Mistral-7B | 32k | Jiang et al. [2023] | `mistralai/Mistral-7B-Instruct-v0.2` |
| FILM-7B | 32k | An et al. [2024] | `In2Training/FILM-7B` |
| Llama-2-7B | 32k | TogetherAI [2024] | `togethercomputer/LLaMA-2-7B-32K` |
| Llama-2-7B | 80k | Fu et al. [2024] | `yaofu/llama-2-7b-80k` |
| Llama-3-8B | 1048k | GradientAI [2024b] | `gradientai/Llama-3-8B-Instruct-Gradient-1048k` |
| Llama-3-70B | 1048k | GradientAI [2024a] | `gradientai/Llama-3-70B-Instruct-Gradient-1048k` |
| Llama-3.1-70B | 128k | Dubey et al. [2024] | `meta-llama/Llama-3.1-70B` |
| Yi-6B | 200k | 01.AI et al. [2024] | `01-ai/Yi-6B-200K` |
| Yi-9B | 200k | 01.AI et al. [2024] | `01-ai/Yi-9B-200K` |
| Yi-34B | 200k | 01.AI et al. [2024] | `01-ai/Yi-34B-200K` |
| Cmd-R-35B | 128k | Cohere for AI [2024] | `CohereForAI/c4ai-command-r-v01` |

## A.2  Tasks

We select 64 classification tasks from huggingface datasets [Lhoest et al., 2021], following the desiderata listed in §4. We provide their references and huggingface identifiers in Table 6. For further use, readers should refer to the licenses of the original datasets.

## A.3  Details on "Pass Rate"

**"Pass Rate" Definition.**    In §3.2, we introduced "pass rate" as the core evaluation metric in Task Haystack. Here we further explain its definition and our considerations when designing this metric. As illustrated in Fig. 6, we first obtain two groups of 5 different performance metrics (*e.g.*, accuracies), one group using Lifelong ICL prompts, and one group using Single-task ICL prompts; we then use two-sided t-test to examine whether the two groups are significantly different. More specifically, we use `scipy.stats.ttest_rel` that returns the t-statistic and p-value for the test, and we consider tests with $p < 0.05$ as significant differences. We choose to use *two-sided* tests to account for potential positive transfers that may arise in the Lifelong ICL setting (§5.3).

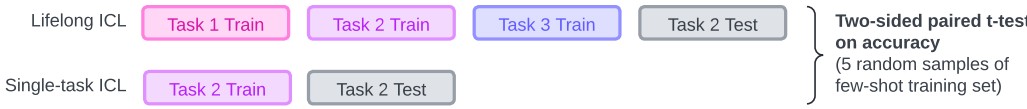

Figure 6: **Definition of "Pass Rate" in Task Haystack.** The model "passes" when the performance of Lifelong ICL is not significantly worse than the Single-task ICL baseline.

**"Pass Rate" Limitations.**    When computing and aggregating pass rates for multiple tasks, we are effectively performing multiple t-tests in parallel, which might increase the risk of Type I errors. We acknowledge this as a limitation of our approach. Following reviewer feedback, we have tried Bonferroni Correction and Benjamini-Hochberg Correction to account for this. However, these methods lead to new challenges. The first method significantly increases the risk of Type II errors and may lead to overestimated pass rates. The second method may lead to unfair comparison across models. Given these concerns, we decide to maintain the current design of pass rates. While this may affect the quantitative results, the qualitative conclusions in the paper are not expected to change.

Table 6: Tasks included in Task Haystack.

| Name | Reference | Huggingface Identifier | License |
|---|---|---|---|
| acl-arc | Bird et al. [2008] | hrithikpiyush/acl-arc | Apache 2.0 |
| ag-news | Zhang et al. [2015] | fancyzhx/ag_news | Unspecified |
| amazon-counterfactual-en | O'Neill et al. [2021] | SetFit/amazon_counterfactual_en | CC BY-NC 4.0 |
| amazon-massive-scenario | FitzGerald et al. [2023] | SetFit/amazon_massive_scenario_en-US | Apache 2.0 |
| app-reviews | Grano et al. [2017] | sealuzh/app_reviews | Unspecified |
| babi-nli | Weston et al. [2015] | tasksource/babi_nli | BSD |
| beaver-tails | Ji et al. [2023] | PKU-Alignment/BeaverTails | CC BY-NC 4.0 |
| boolq | Clark et al. [2019] | google/boolq | CC BY-SA 3.0 |
| brag-action | Choi et al. [2023] | Blablablab/SOCKET | CC BY 4.0 |
| cb | de Marneffe et al. [2019] | aps/super_glue | Unspecified |
| circa | Louis et al. [2020] | google-research-datasets/circa | CC BY 4.0 |
| clickbait | Chakraborty et al. [2016] | marksverdhei/clickbait_title_classification | MIT |
| climate-commitments-actions | Bingler et al. [2024] | climatebert/climate_commitments_actions | CC-BY-NC-SA 4.0 |
| climate-fever | Diggelmann et al. [2020] | tdiggelm/climate_fever | Unspecified |
| climate-sentiment | Bingler et al. [2024] | climatebert/climate_sentiment | CC BY-NC-SA 4.0 |
| cola | Warstadt et al. [2019] | nyu-mll/glue | Other |
| copa | Roemmele et al. [2011] | aps/super_glue | BSD 2-Clause |
| covid-fake-news | Patwa et al. [2021] | nanyy1025/covid_fake_news | Unspecified |
| dbpedia14 | Zhang et al. [2015] | fancyzhx/dbpedia_14 | CC BY-SA 3.0 |
| disaster-repsonse-message | | community-datasets/disaster_response_messages | Unspecified |
| emo | Chatterjee et al. [2019] | SemEvalWorkshop/emo | Unspecified |
| emotion | Saravia et al. [2018] | dair-ai/emotion | Unspecified |
| environmental-claims | Webersinke et al. [2021] | climatebert/environmental_claims | CC BY-NC-SA 4.0 |
| ethos | Mollas et al. [2022] | iamollas/ethos | AGPL 3.0 |
| fever | Thorne et al. [2018] | fever/fever | CC BY-SA 3.0, GPL 3.0 |
| financial-phrasebank | Malo et al. [2013] | takala/financial_phrasebank | CC BY-NC-SA 3.0 |
| function-of-decision-section | Guha et al. [2023] | nguha/legalbench | CC BY 4.0 |
| hate-speech18 | de Gibert et al. [2018] | odegiber/hate_speech18 | CC BY-SA 3.0 |
| health-fact | Kotonya and Toni [2020] | ImperialCollegeLondon/health_fact | MIT |
| i2d2 | Bhagavatula et al. [2023] | tasksource/I2D2 | Apache 2.0 |
| imdb | Maas et al. [2011] | stanfordnlp/imdb | Unspecified |
| insincere-questions | Ellis et al. [2018] | SetFit/insincere-questions | Unspecified |
| is-humor | Meaney et al. [2021] | Blablablab/SOCKET | CC BY 4.0 |
| jailbreak-classification | | jackhhao/jailbreak-classification | Apache 2.0 |
| lexical-rc-cogalexv | Santus et al. [2016a] | relbert/lexical_relation_classification | Unspecified |
| lexical-rc-root09 | Santus et al. [2016b] | relbert/lexical_relation_classification | Unspecified |
| liar | Wang [2017] | ucsbnlp/liar | Unspecified |
| limit | Manotas et al. [2020] | IBM/limit | CC BY-SA 4.0 |
| logical-fallacy-detection | Srivastava et al. [2023] | tasksource/bigbench | Apache 2.0 |
| medical-question-pairs | McCreery et al. [2020] | curaihealth/medical_questions_pairs | Unspecified |
| metaphor-boolean | Bizzoni and Lappin [2018] | tasksource/bigbench | Apache 2.0 |
| mnli | Williams et al. [2018] | nyu-mll/multi_nli | CC BY 3.0, CC BY-SA 3.0, MIT, Other |
| mrpc | Dolan and Brockett [2005] | nyu-mll/glue | Unspecified |
| news-data | | okite97/news-data | AFL 3.0 |
| poem-sentiment | Sheng and Uthus [2020] | google-research-datasets/poem_sentiment | CC BY 4.0 |
| pragmeval-emergent | Ferreira and Vlachos [2016] | sileod/pragmeval | Unspecified |
| pragmeval-sarcasm | Oraby et al. [2016] | sileod/pragmeval | Unspecified |
| pragmeval-verifiability | Park and Cardie [2014] | sileod/pragmeval | Unspecified |
| prosocial-dialog | Kim et al. [2022] | allenai/prosocial-dialog | CC BY 4.0 |
| pun-detection | Miller et al. [2017] | frostymelonade/SemEval2017-task7-pun-detection | CC BY NC |
| qnli | Rajpurkar et al. [2016] | nyu-mll/glue | CC BY-SA 4.0 |
| qqp | Iyer et al. [2016] | nyu-mll/glue | Others |
| rct20k | Dernoncourt and Lee [2017] | armanc/pubmed-rct20k | Unspecified |
| rotten-tomatoes | Pang and Lee [2005] | cornell-movie-review-data/rotten_tomatoes | Unspecified |
| rte | Wang et al. [2018] | nyu-mll/glue | Unspecified |
| sara-entailment | Holzenberger et al. [2020] | nguha/legalbench | MIT |
| scierc | Luan et al. [2018] | hrithikpiyush/scierc | Unspecified |
| semeval-absa-laptop | Pontiki et al. [2015] | jakartaresearch/semeval-absa | CC BY 4.0 |
| semeval-absa-restaurant | Pontiki et al. [2015] | jakartaresearch/semeval-absa | CC BY 4.0 |
| senteval-cr | Hu and Liu [2004] | SetFit/SentEval-CR | BSD |
| senteval-subj | Pang and Lee [2004] | SetFit/subj | BSD |
| sick | Marelli et al. [2014] | RobZamp/sick | CC BY-NC-SA 3.0 |
| silicon-dyda-da | Chapuis et al. [2020] | eusip/silicone | CC BY-SA 4.0 |
| sms-spam | Almeida et al. [2011] | ucirvine/sms_spam | Unspecified |
| sst2 | Socher et al. [2013] | stanfordnlp/sst2 | Unspecified |
| sst5 | Socher et al. [2013] | SetFit/sst5 | Unspecified |
| stance-abortion | Mohammad et al. [2016] | cardiffnlp/tweet_eval | Unspecified |
| stance-feminist | Mohammad et al. [2016] | cardiffnlp/tweet_eval | Unspecified |
| student-question-categories | Biswal [2020] | SetFit/student-question-categories | CC0 |
| tcfd-recommendations | Bingler et al. [2024] | climatebert/tcfd_recommendations | CC BY-NC-SA 4.0 |
| this-is-not-a-dataset | García-Ferrero et al. [2023] | HiTZ/This-is-not-a-dataset | Apache 2.0 |
| toxic-conversations | cjadams et al. [2019] | SetFit/toxic_conversations | CC0 |
| trec | Li and Roth [2002] | CogComp/trec | Unspecified |
| vitaminc | Schuster et al. [2021] | tals/vitaminc | CC BY-SA 3.0 |
| wic | Pilehvar and Camacho-Collados [2019] | aps/super_glue | CC BY-NC 4.0 |
| wiki-hades | Liu et al. [2022b] | tasksource/wiki-hades | MIT |
| wiki-qa | Yang et al. [2015] | microsoft/wiki_qa | Other |
| wnli | Levesque et al. [2011] | nyu-mll/glue | Unspecified |
| wsc | Kocijan et al. [2019] | aps/super_glue | CC BY 4.0 |
| yahoo-answers-topics | Zhang et al. [2015] | community-datasets/yahoo_answers_topics | Unspecified |

## A.4 Implementation and Engineering Details

**Data Preprocessing.** For each task, the authors manually wrote two task instructions and a task template for in-context learning. In the following we provide one example for the task of `ag_news`. We ensure that all options have distinct starting tokens when writing the task template, so that the inference can be done with rank classification [Liu et al., 2022a].

```
1  {
2      "name": "ag_news",
3      "task_type": "classification",
4      "options": ["World", "Sports", "Business", "Technology"],
5      "instruction": "Classify the news article into World, Sports, Business or
           Technology.",
6      "instruction_2": "Determine which category best fits the news article: Sports,
           Technology, Business, or World.",
7      "demonstration_prompt": "Article: {text}\nAnswer: {label}",
8      "inference_prompt": "Article: {text}\nAnswer:"
9  }
```

To create the few-shot training sets, we randomly sampled five subsets from the original training dataset for in-context learning, each containing at least 16 examples per class. We sub-sample 100 instances from the original development set (or test set when the development set is not provided) to form our test set. Our preprocessing scripts are included in the released code ⬡ INK-USC/Lifelong-ICL.

**LLM Inference.** We apply rank classification [Liu et al., 2022a] in all our experiments. Specifically, we query the LM with the prompt and obtain the top 100 predictions for the next token. We then cross-reference this list with the list of the first token of all possible options. We use the prefix caching technique in vLLM [Kwon et al., 2023] which significantly improves the inference speed.

We did *not* use model-specific prompts (*e.g.*, chat template, special tokens, instructions optimized for a specific model). This decision reduces experiment complexity and is reasonable because (1) we expect a model optimized for chat to still be able to perform ICL as a text-token prediction task; (2) the special tokens (*e.g.*, `<|user|>`, `[INST]`) may create tokenization inconsistencies in in-context learning (*e.g.*, `World` and `_World` may be two different tokens in the vocabulary); (3) Task Haystack is based on a comparative assumption, making absolute accuracies less important.

**Inference Costs.** Running a 64-task, 2-shot Task Haystack experiment with a 7B model on one A6000 GPU takes around 20 hours. Running a 16-task, 8-shot Task Haystack experiment with a 7B model on one A6000 GPU takes around 8 hours. For 34B and 70B models, we use four A6000 GPUs. For evaluations with OpenAI models, we use the Batched API.[8] All experiments using OpenAI models (Table 2 bottom rows; Fig 21-22) incur a total cost of about $8,000 at the time of writing.

# B  Additional Results

## B.1  Scale-Task Experiments

In Table 7, we report the results in the Scale-Task setting, where we fix the number of shots per class $n_{shot}$ to be 2, and experiment with $n_{task} \in \{8, 16, 24, 32, 40, 48, 56, 64\}$.[9] We noticed that the overall pass rates are higher than those in the Scale-Shot setting (Table 2), potentially due to a smaller value of $n_{shot}$. However, long-context models still struggle in this setting: in 41 out of 44 cases in Table 7, the overall pass rates drop below 90%. The three cases achieving pass rates above 90% use the Llama2-7B (80k) model, an outlier model discussed in §5.1.

## B.2  Task Subset with Permissive Licenses

To ensure responsible data use, in this section we introduce a new subset of 16 tasks (Table 8), each with a permissive license. We recommend that users of Task Haystack refer to this subset for future benchmarking and analysis. We have conducted evaluations on selected models

---

[8]https://platform.openai.com/docs/guides/batch

[9]Since each column in the table uses a different set of tasks, accuracies and pass rates from different columns are not directly comparable.

Table 7: **Main Results: Fixing 2 Shots, Scaling the Number of Tasks (§B.1).** See the caption of Table 2 for the explanations of the table headers.

| Model | 8 tasks (4k) | | | 16 tasks (8k) | | | 32 tasks (15k) | | | 64 tasks (25k) | | |
|---|---|---|---|---|---|---|---|---|---|---|---|---|
| | S-acc | L-acc | pass | S-acc | L-acc | pass | S-acc | L-acc | pass | S-acc | L-acc | pass |
| Mistral-7B (32k) | 76.4 | 78.9 | 80.0 | 77.6 | 74.6 | 73.8 | 72.7 | 71.1 | 72.5 | 70.6 | 69.3 | 75.6 |
| FILM-7B (32k) | 79.1 | 77.1 | 87.5 | 79.1 | 75.1 | 77.5 | 73.3 | 72.0 | 88.1 | 70.6 | 69.7 | 75.3 |
| Llama-2-7B (32k) | 70.1 | 60.7 | 65.0 | 72.8 | 64.5 | 53.8 | 70.6 | 64.5 | 59.4 | 67.1 | 61.2 | 63.1 |
| Llama-2-7B (80k) | 49.9 | 58.5 | 97.5 | 49.8 | 60.2 | 100.0 | 49.5 | 58.3 | 91.2 | 48.6 | 52.0 | 89.7 |
| Llama-3-8B (1048k) | 68.3 | 65.4 | 75.0 | 70.0 | 69.1 | 76.2 | 67.4 | 65.1 | 75.6 | 66.4 | 65.7 | 81.2 |
| Llama-3-70B (1048k) | 77.1 | 73.8 | 45.0 | 79.0 | 74.4 | 50.0 | 76.0 | 61.6 | 59.4 | 74.1 | 70.4 | 70.3 |
| Llama-3.1-70B (128k) | 81.5 | 78.2 | 77.5 | 82.8 | 81.1 | 76.2 | 79.2 | 78.0 | 72.5 | 76.9 | 75.6 | 75.3 |
| Yi-6B (200k) | 72.0 | 54.4 | 50.0 | 73.0 | 58.6 | 51.2 | 68.4 | 59.2 | 63.7 | 63.7 | 55.7 | 65.6 |
| Yi-9B (200k) | 78.6 | 73.4 | 62.5 | 77.7 | 72.9 | 71.2 | 75.5 | 70.3 | 61.3 | 70.2 | 66.8 | 61.3 |
| Yi-34B (200k) | 66.1 | 70.7 | 87.5 | 74.1 | 72.4 | 60.0 | 74.0 | 69.7 | 63.1 | 71.5 | 68.2 | 59.4 |
| Cmd-R-35B (128k) | 71.2 | 75.2 | 82.5 | 75.3 | 75.5 | 61.3 | 71.2 | 72.5 | 73.1 | 70.3 | 70.6 | 77.2 |

Table 8: **Subset of 16 Tasks with Permissive Licenses (§B.2).**

| metaphor-boolean | fever | function-of-decision-section | climate-commitments-actions |
|---|---|---|---|
| amazon-massive-scenario | silicone-dyda-da | brag-action | student-question-categories |
| acl-arc | wic | semeval-absa-laptop | senteval-cr |
| dbpedia14 | wiki-hades | environmental-claims | babi-nli |

Table 9: **Additional Results on 16 Tasks with Permissive Licenses (§B.2). Fixing 16 Tasks, Scaling the Number of Shots.** See caption of Table 2 for the explanations of the table headers. "-" indicates that the prompt exceeds the maximum context length.

| Model | 1-shot (4k) | | | 2-shot (8k) | | | 4-shot (16k) | | | 8-shot (32k) | | |
|---|---|---|---|---|---|---|---|---|---|---|---|---|
| | S-acc | L-acc | pass | S-acc | L-acc | pass | S-acc | L-acc | pass | S-acc | L-acc | pass |
| Mistral-7B (32k) | 67.5 | 68.9 | 95.0 | 70.7 | 69.4 | 70.0 | 72.3 | 69.7 | 52.5 | - | - | - |
| FILM-7B (32k) | 68.9 | 71.1 | 91.2 | 71.4 | 71.9 | 87.5 | 72.7 | 72.4 | 73.8 | - | - | - |
| Llama-3.1-70B (128k) | 73.8 | 74.2 | 83.8 | 75.0 | 75.1 | 88.7 | 76.6 | 75.7 | 77.5 | 77.5 | 75.6 | 73.8 |
| Llama-3.2-1B (128k) | 50.6 | 43.9 | 71.3 | 52.2 | 44.6 | 68.8 | 57.9 | 46.3 | 62.5 | 58.4 | 45.7 | 68.7 |
| Llama-3.2-3B (128k) | 59.2 | 62.7 | 93.8 | 60.8 | 63.3 | 62.5 | 64.0 | 64.2 | 68.8 | 63.7 | 64.4 | 66.2 |
| Gemini-1.5-Flash (128k) | 74.6 | 76.5 | 100.0 | 73.1 | 76.8 | 97.5 | 70.8 | 77.2 | 91.2 | 68.3 | 77.2 | 87.5 |
| GPT-4o-mini (128k) | 73.0 | 72.2 | 80.0 | 74.0 | 72.8 | 82.5 | 73.0 | 73.7 | 80.0 | 72.3 | 73.8 | 85.0 |

with this subset. We also evaluated recent models that were released after the submission date, including `meta-llama/Llama-3.2-1B-Instruct`, `meta-llama/Llama-3.2-3B-Instruct` and `gpt-4o-mini-2024-07-18`. The results are presented in Table 9.

## B.3 Controlled Analysis

In Fig. 7-8, we repeat the controlled experiments in Fig. 4-5, using N-task=64, N-shot=2 instead of N-task=16, N-shot=4. Our observations are generally consistent with those in Fig. 4-5. One exception is Mistral-7B (32k) experiments in Fig. 7, where the model achieves comparable accuracies in Recall, Replay and Remove. We attribute this to the usage of a smaller N-shot value compared to Fig. 4.

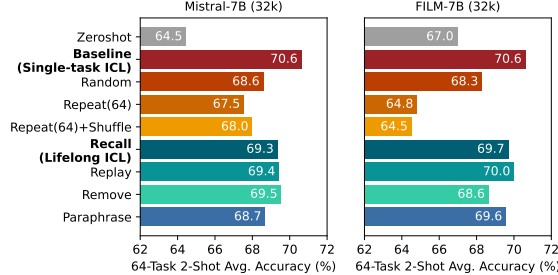

Figure 7: **Controlled Experiments.** We repeat the experiments in Fig. 4 with N-task=64 and N-shot=2. The gaps between control settings are smaller, possibly due to a smaller value of N-shot.

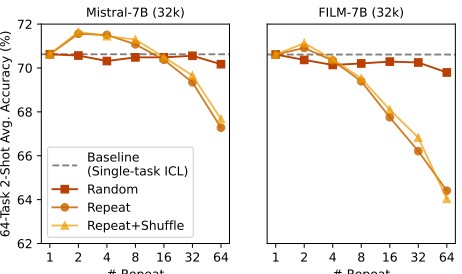

Figure 8: **"Multi-epoch" ICL.** We repeat the experiments in Fig. 5 with N-task=64 and N-shot=2. The increase-then-decrease phenomenon is more evident in this scenario.

## B.4 Comparing Base and Instruct Models

Previously in Table 2, we experimented with the base version of Llama-3.1-70B model (*i.e.*, `meta-llama/Llama-3.1-70B`), and identified it as the most capable open-weight model among those evaluated. Here in Table 10, we additionally report results using its instruct version (*i.e.*, `meta-llama/Llama-3.1-70B-Instruct`). Compared to the base model, the instruct model demonstrates improvements in S-acc across all settings, with L-acc remaining comparable or slightly improved. The increased S-acc raises the reference threshold, resulting in lower overall pass rates.

One possible explanation is that instruction tuning and subsequent post-training processes, which primarily involve shorter texts and conversational data, may encourage the model to focus more on the beginning of the context. This shift could potentially compromise its ability to handle long contexts. Given this observation, we believe Lifelong ICL and Task Haystack can also serve as a tool to monitor the long-context modeling capabilities before and after post-training processes.

Table 10: **Comparing Base and Instruct Versions of Llama-3.1-70B**. We use the 16-task Scale-Shot setting, consistent with Table 2.

| Model | 0-shot | 1-shot (4k) | | | 2-shot (8k) | | | 4-shot (16k) | | | 8-shot (32k) | | |
|---|---|---|---|---|---|---|---|---|---|---|---|---|---|
| | S-acc | S-acc | L-acc | pass | S-acc | L-acc | pass | S-acc | L-acc | pass | S-acc | L-acc | pass |
| Llama-3.1-70B (128k) | 58.8 | 81.7 | 81.2 | 80.0 | 82.8 | 81.1 | 76.2 | 84.6 | 82.4 | 83.8 | 85.2 | 83.3 | 80.0 |
| Llama-3.1-70B-Inst. (128k) | 78.4 | 84.0 | 82.2 | 72.5 | 84.9 | 82.4 | 63.7 | 85.8 | 82.7 | 72.5 | 86.9 | 83.2 | 67.5 |

## B.5 Original NIAH Experiments

We experiment with the original needle-in-a-haystack evaluation [Kamradt, 2023] to provide reference. We use the question "What is the best thing to do in San Francisco?" and the needle "The best thing to do in San Francisco is eat a sandwich and sit in Dolores Park on a sunny day." The metric is the token-level recall of the model's response. We report the pass rates in Table 11 and visualize the results in the first column of figures in Fig. 9-18.

## B.6 Interleaving Examples from Multiple Tasks

The complexity of Lifelong ICL can be further increased by interleaving in-context learning examples from multiple tasks, analogous to a multi-needle NIAH challenge [Hsieh et al., 2024]. We conducted preliminary experiments of this setting, where each ICL example is paired with its task instruction before itself, and ICL examples of different tasks are streamed in a random order. The results are presented in Table 12. We observe that accuracies in this setting (M-acc) are generally lower than those in the non-interleaving Lifelong ICL setting (L-acc), confirming that interleaving examples adds more challenges in locating relevant context. We leave further investigation as future work.

Table 11: **Results of the original NIAH evaluation (§B.5).**

| Model | Pass (%) | Model | Pass (%) |
|---|---|---|---|
| Mistral-7B (32k) | 95.3 | Llama-3-8B (1048k) | 100.0 |
| FILM-7B (32k) | 100.0 | Yi-6B (200k) | 100.0 |
| Llama-2-7B (32k) | 95.3 | Yi-9B (200k) | 100.0 |
| Llama-2-7B (80k) | 100.0 | Yi-34B (200k) | 100.0 |

Table 12: **Results of Interleaving Examples Across Tasks (§B.6).** Experiments done in the 16-Task, 4-shot setting. "M-acc" indicates results of interleaving examples.

| Model | S-acc | L-acc | M-acc |
|---|---|---|---|
| Mistral-7B (32k) | 78.6 | 74.8 | 72.1 |
| FILM-7B (32k) | 79.6 | 75.4 | 74.7 |

## C Additional Discussion

**Task Haystack can be solved by a RAG baseline.** To examine whether Task Haystack can be addressed by retrieval-augmented generation (RAG) methods, we implemented a simple RAG baseline. We used an off-the-shelf retriever [Zhang, 2024] to select one prompt from all task-specific ICL prompts (*i.e.*, $p_{a_1}, p_{a_2}, ..., p_{a_n}$ as defined in §3). The selected prompt was then prepended before the instruction of

Table 13: **Results of RAG Baseline (§C).** Experiments done with 16-Task, 4-Shot. "RAG-acc" indicates the RAG baseline results. Single-task ICL (S-acc) can be seen as an oracle setting with perfect retrieval accuracy.

| Model | S-acc | L-acc | RAG-acc |
|---|---|---|---|
| Mistral-7B (32k) | 78.6 | 74.8 | 79.0 |
| FILM-7B (32k) | 79.6 | 75.4 | 80.1 |

the test task. We report the results in Table 13. As expected, Task Haystack, being a retrieval-style task, can be solved by this RAG baseline.

**Task Haystack is still meaningful for long-context LM evaluation.** Although Task Haystack can be solved by a RAG baseline, we believe Task Haystack—and retrieval-style tasks more broadly—remain valuable for evaluating long-context models. **(1)** One advantage of retrieval-style tasks is that it's much more controllable for ablations. This allows us to carefully investigate whether these long-context LMs behave robustly and as expected. **(2)** As pointed out by Lee et al. [2024], long-context models have certain benefits over RAG methods, including having a simpler pipeline, better handling of multi-hop queries and mitigating cascading errors. HELMET [Yen et al., 2024], a recently-released long-context benchmark, also incorporates retrieval-style and retrieval-augmented generation tasks.

**Task Haystack adds to the axis of *contextual understanding* in long-context LM evaluation.** Goldman et al. [2024] introduce a taxonomy of long-context LM evaluation with two axes: (1) *diffusion*: how hard it is to find and extract the necessary information, and (2) *scope*: how long the necessary information is. In the view of this taxonomy, Task Haystack is having low diffusion (it's not hard to find relevant information) and small scope (the relevant information is short, only a few ICL examples), similar to the original NIAH. However, Task Haystack is also more challenging than the original NIAH partly due to requiring contextual understanding (or "implicit aggregations" as briefly mentioned in Goldman et al. [2024]) of the context. We believe it may be helpful to add a third axis of "contextual understanding" or "implicit aggregation" to the taxonomy, and Task Haystack can be seen as making progress in this third axis.

# D    NIAH-style Visualizations

In Fig. 9-19, we present detailed Task Haystack results for ten open-weight models. In Fig. 20-22 we present results for Gemini-1.5-Flash, GPT-3.5-Turbo and GPT-4o.

Fig. 9-18 each contains three subfigures: On the left side, we illustrate the results of the original needle-in-a-haystack evaluation [Kamradt, 2023], described in §B.5. In the middle, we visualize the results of the Scale-Shot setting. On the right side, we visualize the results of the Scale-Task setting. See §4 for the details of the two scaling settings.

In each subfigure, the x-axis represents the input context length, and the y-axis represents the depth of the key information (*i.e.*, "needle"). In figures visualizing Task Haystack results, a red cell represents that the model is failing the test (*i.e.*, Lifelong ICL being significantly worse than Single-task ICL) and a blue cell represents that the model is excelling the test (*i.e.*, Lifelong ICL being significantly better than Single-task ICL, potentially due to positive transfer).

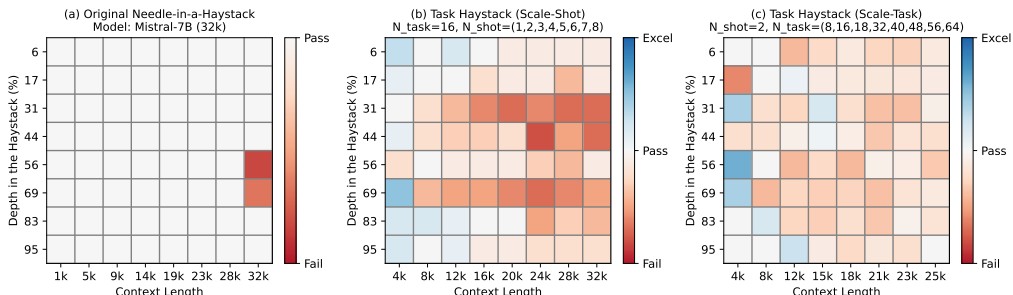

Figure 9: Task Haystack Results on Mistral-7B (32k).

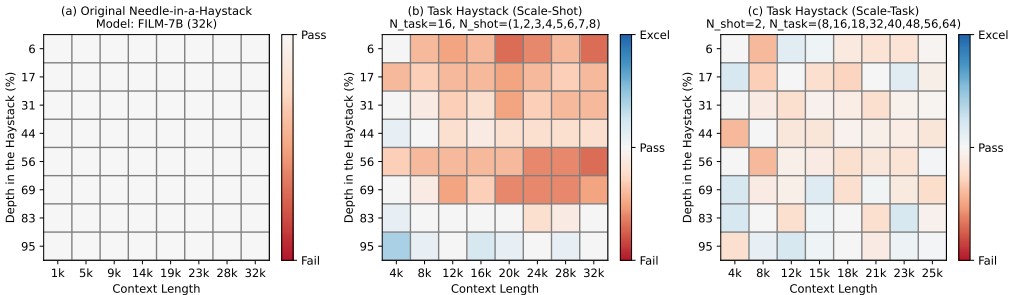

Figure 10: Task Haystack Results on FILM-7B (32k).

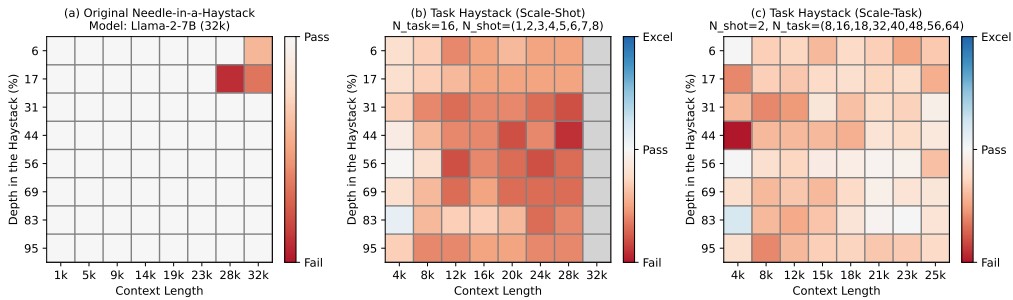

Figure 11: Task Haystack Results on Llama-2-7B (32k).

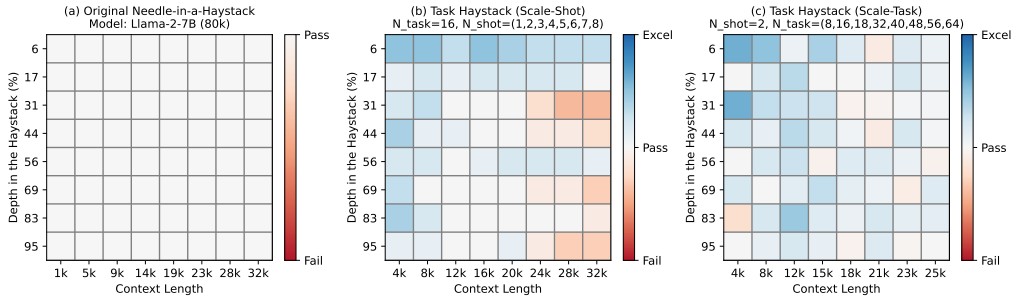

Figure 12: Task Haystack Results on Llama-2-7B (80k).

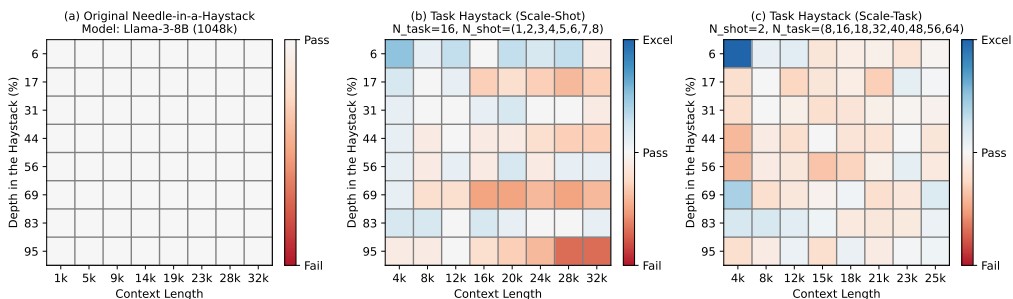

Figure 13: Task Haystack Results on Llama-3-8B (1048k).

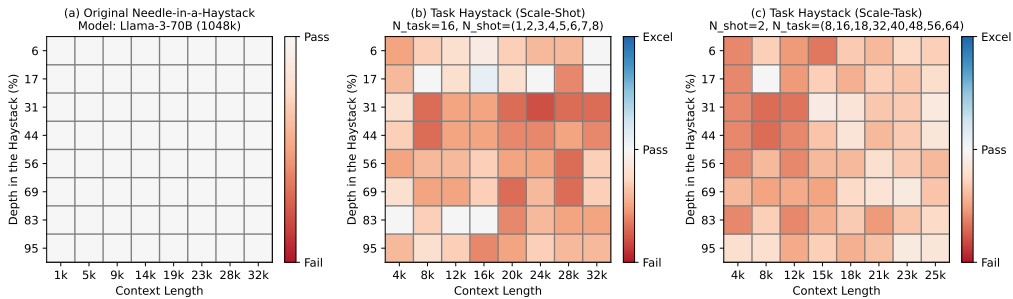

Figure 14: Task Haystack Results on Llama-3-70B (1048k).

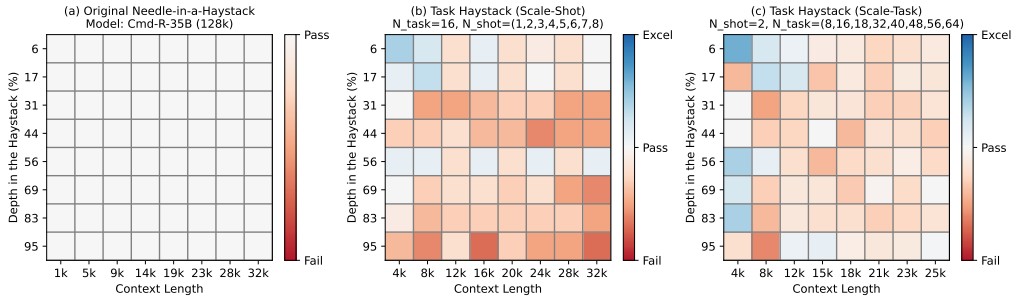

Figure 15: Task Haystack Results on Cmd-R-35B (128k).

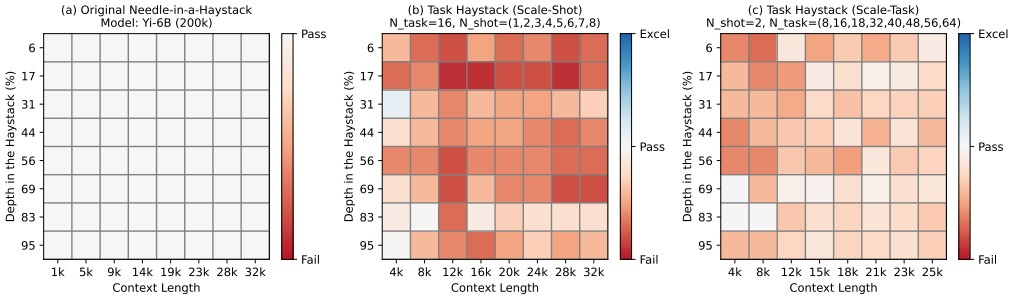

Figure 16: Task Haystack Results on Yi-6B (200k).

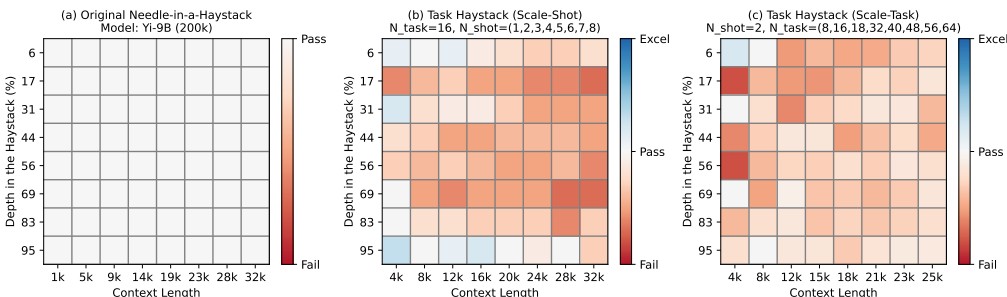

Figure 17: Task Haystack Results on Yi-9B (200k).

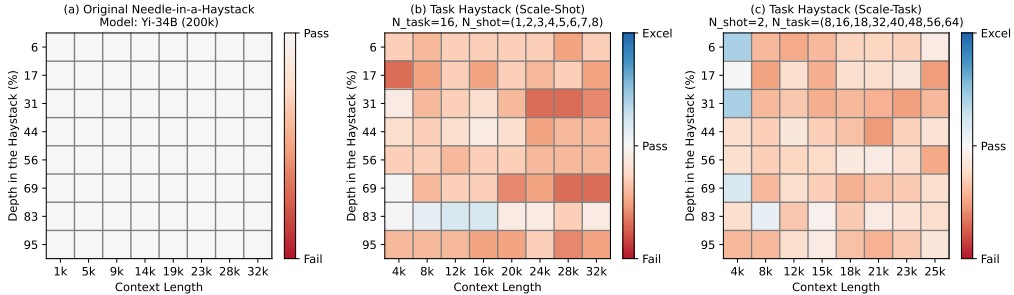

Figure 18: Task Haystack Results on Yi-34B (200k).

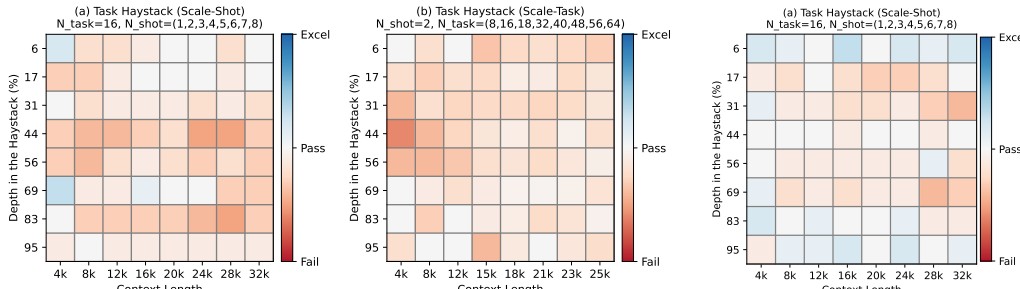

Figure 19: Task Haystack Results on Llama-3.1-70B (128k). Figure 20: Task Haystack Results on Gemini-1.5-Flash (128k).

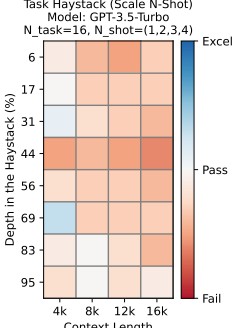

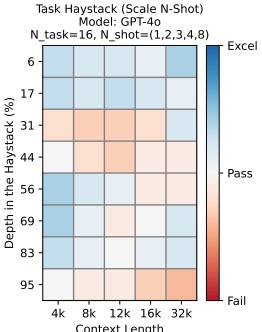

Figure 21: Task Haystack Results on GPT-3.5-Turbo (16k). Due to budget limits we only experiment with the Scale-Shot setting.

Figure 22: Task Haystack Results on GPT-4o (128k). Due to budget limits we only experiment with the Scale-Shot setting and skipped N-shot=5,6,7.

# E Fine-grained Diagnostic Reports

Task Haystack inherits the controllability benefits of the original needle-in-a-haystack test [Kamradt, 2023]. It is straight-forward to aggregate results by permutations, context depth, and task, enabling the creation of visualized reports to help identify the vulnerabilities of long-context LMs.

In the following, we provide visualizations of 6 sets of experiments discussed in the main paper and summarize our main findings. The experiment settings include:

- Fig. 24: Mistral-7B (32k), N-Task=16, N-Shot=8.
- Fig. 25: FILM-7B (32k), N-Task=16, N-Shot=8.
- Fig. 26: GPT-3.5-Turbo (16k), N-Task=16, N-Shot=4.
- Fig. 27: GPT-4o (128k), N-Task=16, N-Shot=8.
- Fig. 28: Mistral-7B (32k), N-Task=32, N-Shot=2.
- Fig. 29: Mistral-7B (32k), N-Task=64, N-Shot=2.

## E.1 How to interpret the diagnostic report?

The main body of the diagnostic report is an $n \times n$ matrix, where $n$ is the number of tasks used in the experiments. The x-axis represents the task index in the Lifelong ICL stream of all tasks, while the y-axis represents the task name. If the cell at (index 5, insincere questions) is colored red, it indicates that the task of insincere questions appears at index 5 in one of the five permutations, and the performance when using the Lifelong ICL prompt is significantly worse than when using the single-task ICL prompt, resulting in a test failure in Task Haystack. A white cell suggests no significant differences, and a blue cell suggests that Lifelong ICL outperforms Single-task ICL. Since we run five permutations of tasks in our experiments, the figure is only sparsely colored. A grey cell means "N/A" and indicates that the task does not appear at a specific index in the five sampled permutations.

Below the main matrix, we plot the results according to the five permutations we created. If the cell at (permutation 1, index 5) is colored red, it indicates that the task at index 5 in permutation 1 failed the Task Haystack test. We average each column and each row in the main $n \times n$ matrix to aggregate performance by task and by index, and visualize them at the right or the bottom of the report. This helps to investigate which tasks are more likely to fail (or excel) and to understand which positions in the context window are more vulnerable.

## E.2 Main Findings

**Failing and excelling are highly task-dependent.** In Fig. 23 we plot the histogram of failure/excel rates grouped by tasks, in the experiments with Mistral-7B (32k), N-Task=64, N-Shot=2. The category "Fail (5/5)" achieves the second-highest frequency, suggesting that these tasks are inherently more likely to be influenced (or "forgotten") in Lifelong ICL, regardless of their position in the context. Similarly, the bars for Excel 3/5, 4/5, 5/5 have higher frequencies than Excel 1/5, 2/5, indicating that certain tasks are inherently more likely to benefit from positive transfer compared to others.

**Different models demonstrate different patterns.** In Table 14, we list the names of tasks that *always fail* (*i.e.*, fail in 5 out of the 5 task permutations) and the names of tasks that *often excel* (*i.e.*, excel in more than 3 out of 5 permutations) in Lifelong ICL for various models.

Our findings show little consistency across the different models investigated. For example, the task brag_action often excels with Mistral-7B (32k) but always fails with FILM-7B (32k) and GPT-3.5-Turbo (16k). Similarly, the task insincere_questions also appear in both categories for different models. One hypothesis is that the compared models may have been trained on the tasks we use, thereby influencing their forgetting and transfer behavior. However, due to the lack of transparency regarding the training details of these models, we cannot further investigate this hypothesis. Another hypothesis is that the Lifelong ICL prompt may influence the model's calibration and consequently the final accuracy. We leave the investigation of this hypothesis for future work.

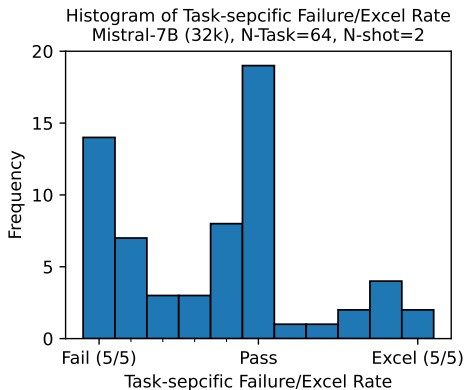

Figure 23: **Histogram Failure/Excel Rate Grouped by Task.** We aggregate the results of Mistral-7B (32k), N-Task=64, N-Shot=2 (Fig. 29).

Table 14: **Notable Tasks By Investigating Task Haystack Results.** We select tasks that always fail for a model (*i.e.*, fail in 5 out of the 5 permutations) and tasks that often excel (*i.e.*, excel in more than 3 out of 5 permutations).

| Model | N-Task | N-Shot | Tasks that always fail (=5/5) | Tasks that often excel (>3/5) |
|---|---|---|---|---|
| Mistral-7B (32k) | 16 | 8 | insincere_questions
news_data | brag_action
wiki_qa |
| FILM-7B (32k) | 16 | 8 | brag_action
emo
insincere_questions | pun_detection |
| GPT-3.5-Turbo (16k) | 16 | 4 | amazon_counterfactual_en
brag_action
this_is_not_a_dataset | - |
| GPT-4o (128k) | 16 | 8 | - | covid_fake_news
insincere_questions
logical_fallacy_detection |

## E.3  Visualizations

### E.3.1  Mistral-7B, N-task=16, N-shot=8

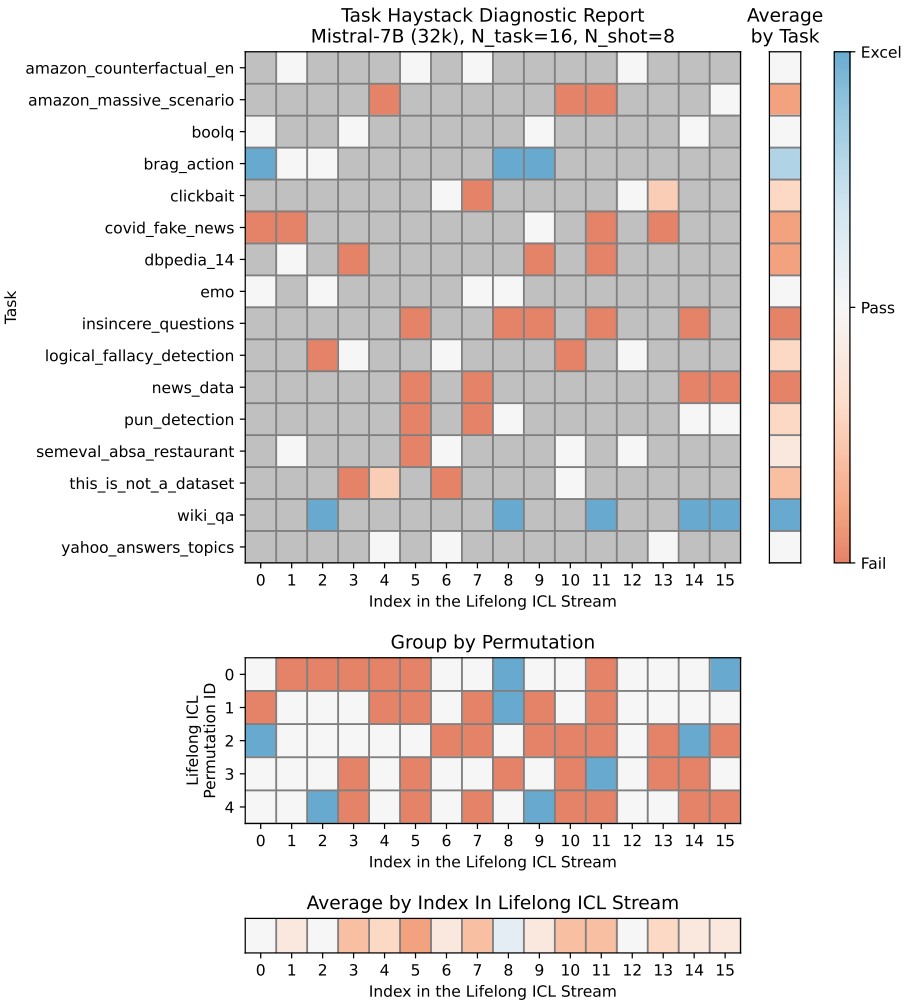

Figure 24: Diagnostic Report on Mistral-7B (32k), N-task=16, N-shot=8. Grey cells indicate that the task does not appear at a given index in the 5 sampled permutations.

## E.3.2 FILM-7B, N-task=16, N-shot=8

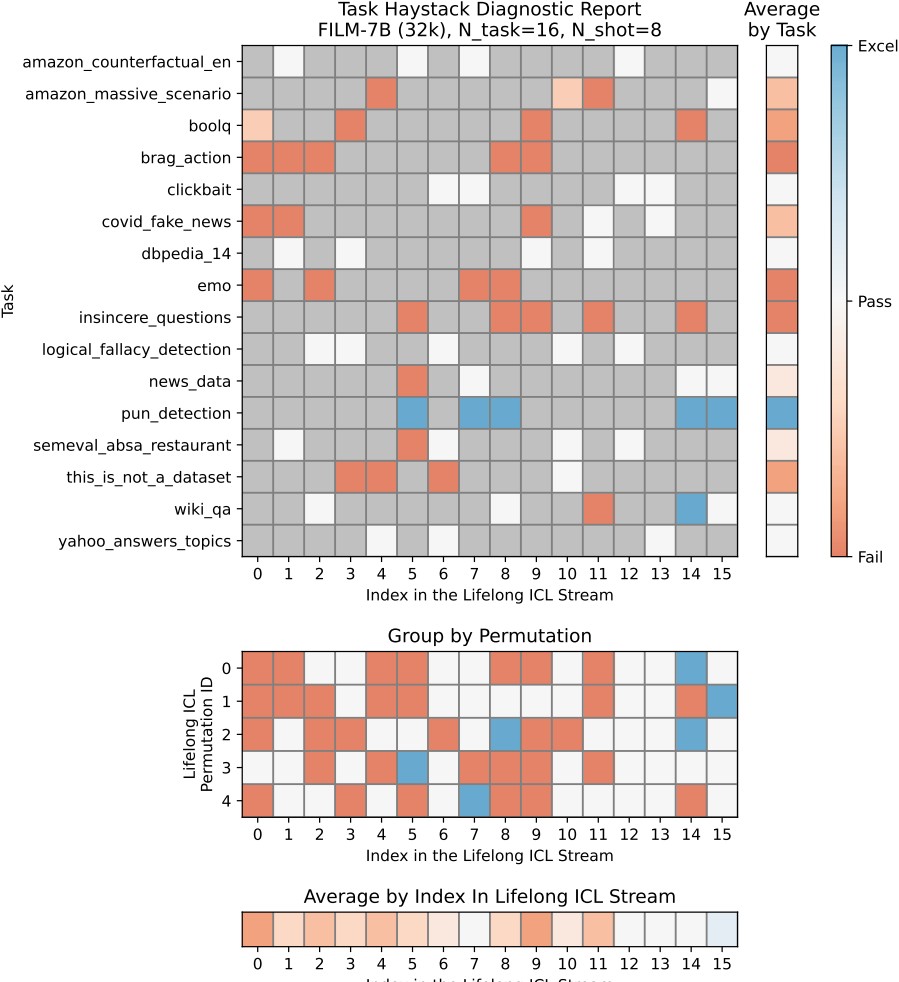

Figure 25: Diagnostic Report on FILM-7B (32k), N-task=16, N-shot=8.

### E.3.3 GPT-3.5-Turbo, N-task=16, N-shot=4

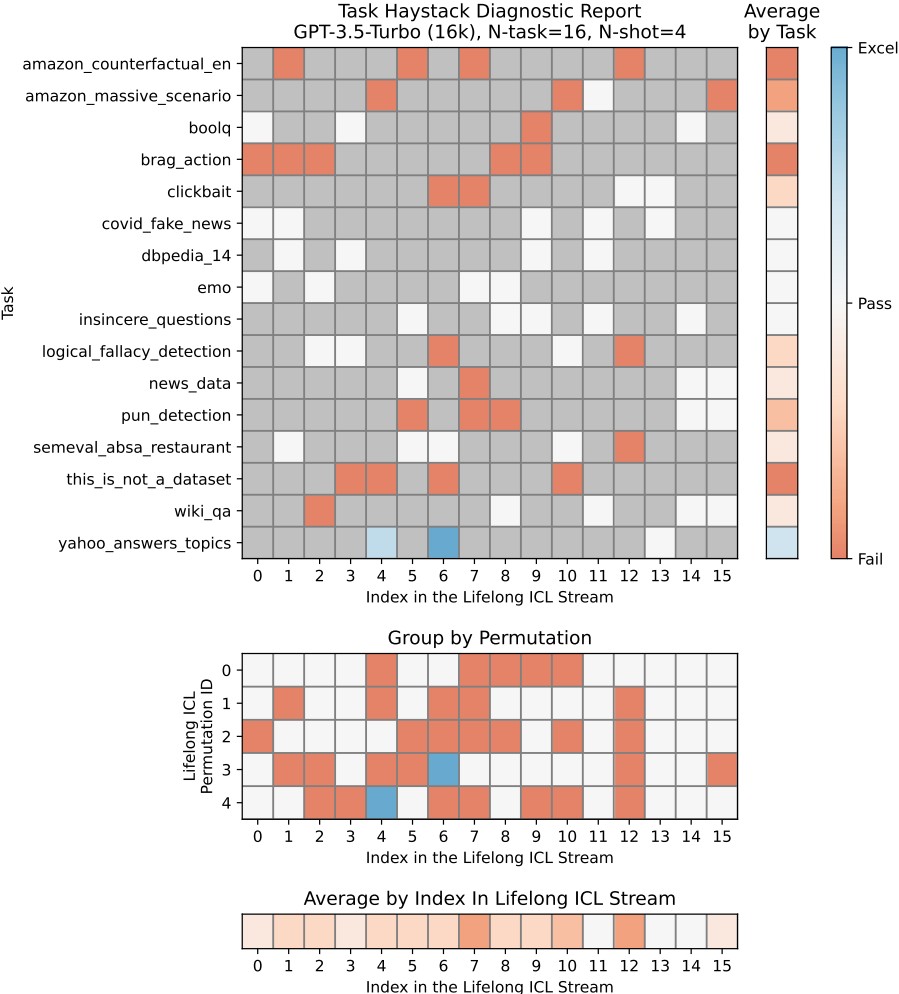

Figure 26: Diagnostic Report on GPT-3.5-Turbo (16k), N-task=16, N-shot=4.

### E.3.4 GPT-4o, N-task=16, N-shot=8

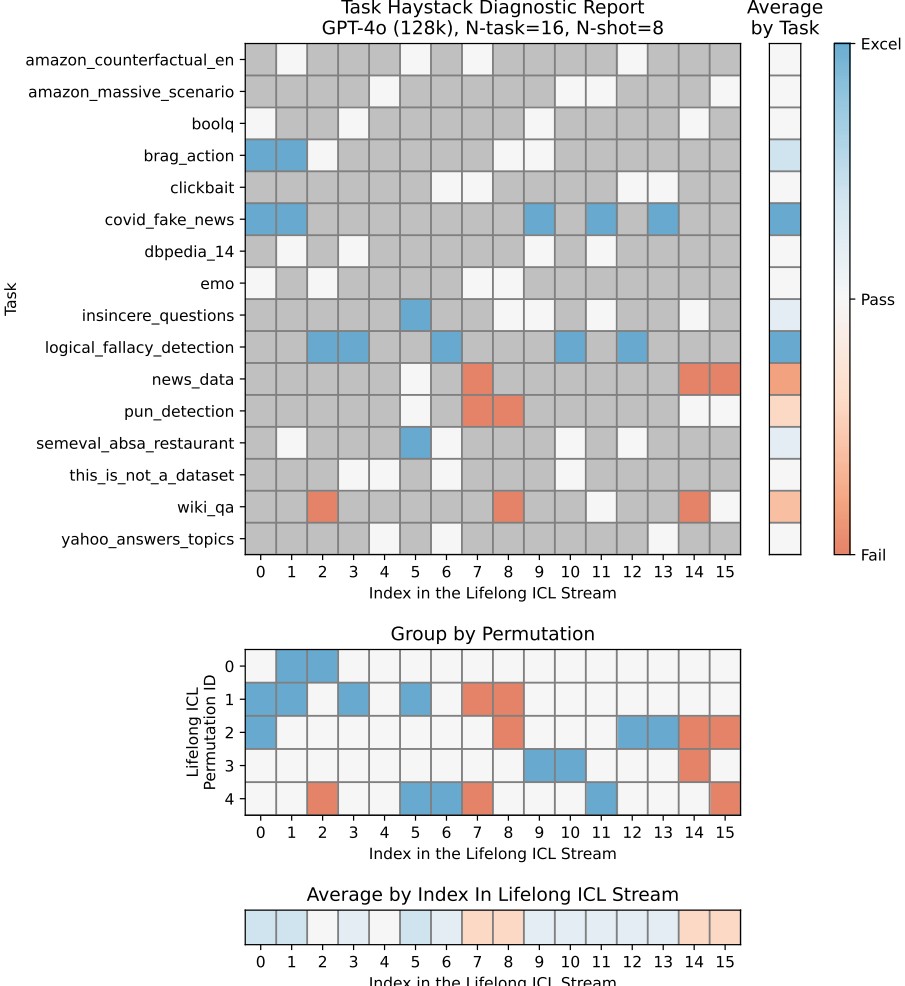

Figure 27: Diagnostic Report on GPT-4o (128k), N-task=16, N-shot=8.

## E.3.5 Mistral-7B, 32-task, 2-shot

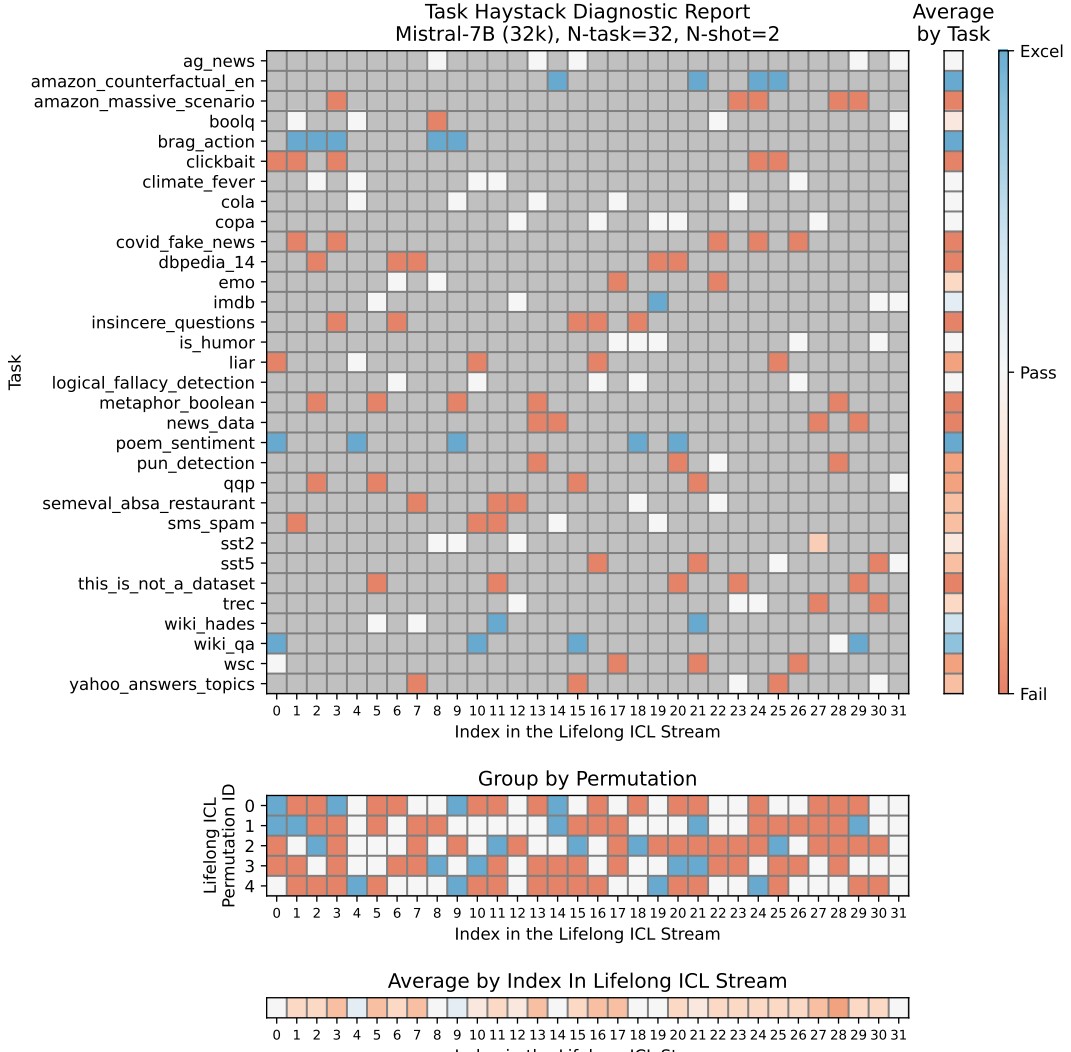

Figure 28: Diagnostic Report on Mistral-7B (32k), N-task=32, N-shot=2.

## E.3.6  Mistral-7B, 64-task, 2-shot

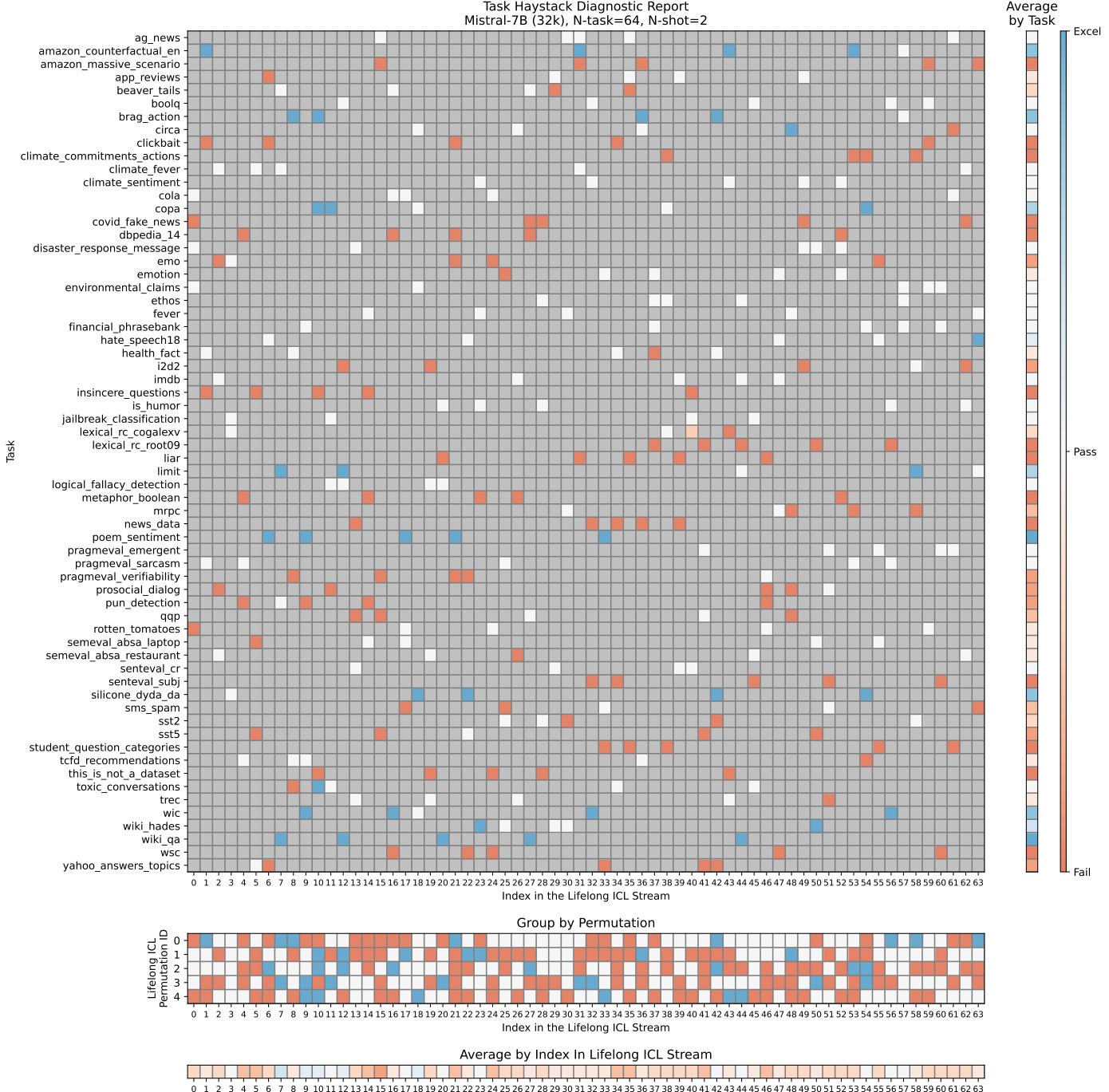

Figure 29: Diagnostic Report on Mistral-7B (32k), N-task=64, N-shot=2.

