# OpenReview forum: "Stress-Testing Long-Context Language Models with Lifelong ICL and Task Haystack"
_NeurIPS.cc/2024/Datasets_and_Benchmarks_Track — NeurIPS 2024 Track Datasets and Benchmarks Poster_

### Official Review · Reviewer_PEya · 2024-07-23

**Rating:** 6
**Confidence:** 3
**Correctness:** Yes
**Clarity:** Yes

**Review:**

Overall the paper is easy to understand.
The research topic and the produced benchmark are timely for rigorously evaluating different long-context LLMs.
Compared to previous work like needle-in-a-haystack [1], the major originality and significance lie in the emphasis on utilizing context in a contextualized manner and long steams of evolving topics and tasks.

**Pros**:

1. The evaluation principles are intuitive and easy to understand.
2. The experiments are comprehensive with detailed analysis.
3. The controlled analysis further reveals some interesting phenomenons when LLMs performing Lifelong ICL

**Cons**:

1. My biggest concern is whether Lifelong ICL is a realistic problem setting. In other words, what would be the motivations and real applications of using long-context LLMs to conduct ICL in this way?

2. Will the low pass rate be improved by a simple RAG module?
That is, given the test task, we retrieve the most relevant tasks from the context and append them just before the test task.
I think the authors should consider this simple RAG baseline.

3. Additionally, could the authors conduct a similar controlled analysis for LLMs that are augmented by a naive RAG module?
I think the community may care more about real long-context issues than those that could not be easily solved by RAG.

[1] https://github.com/gkamradt/LLMTest_NeedleInAHaystack/tree/main

**Strengths:**

See the above list of pros

**Additional Feedback:**

N/A

**Documentation:**

Yes

**Limitations:**

Yes

**Opportunities For Improvement:**

See the above list of cons

**Relation To Prior Work:**

Yes

**Summary And Contributions:**

This paper introduces Task Haystack, a benchmark designed to evaluate long-context LLMs from the perspective of Lifelong ICL.
The primary contributions of this paper are 3 folds:

1. propose a new problem setting Lifelong ICL
2. propose a new benchmark Task Haystack
3. conduct comprehensive experiments on Task Haystack to demonstrate its utilities

---

> ### Author Rebuttal · Authors · 2024-08-17
>
> Dear reviewer PEya,
>
> Thank you for your thoughtful comments! We have added a RAG-baseline per requested, and we hope our response below can address your concerns.
>
> > My biggest concern is whether Lifelong ICL is a realistic problem setting. In other words, what would be the motivations and real applications of using long-context LLMs to conduct ICL in this way?
>
> - To clarify, we consider Lifelong ICL and Task Haystack as a *hybrid* benchmark; we do not claim it to be a fully realistic benchmark at this stage. The synthetic element of it allows us to create very long contexts systematically. In the absence of realistic benchmarks with very long input text, we believe this is a reasonable solution to evaluating long-context LMs. We will adjust our phrasing in the submission to avoid confusion.
>
> - We believe Lifelong ICL is meaningful in several other ways: (1) It serves as a proxy to realistic applications. The task stream in the context approximates the realistic cases where there are shifting and evolving topics in the context. (2) It serves as an exploration step for developing LMs/AI systems that continuously learn and adapt in a gradient-free manner. While it would be more ideal to evaluate LMs with a more realistic task stream (e.g., college student curriculum), we hope our work serves as a meaningful starting point.
>
> > Will the low pass rate be improved by a simple RAG module? That is, given the test task, we retrieve the most relevant tasks from the context and append them just before the test task. I think the authors should consider this simple RAG baseline.
>
> - In Sec. 5.2, our controlled analysis, we have a “Replay” setting that can be seen as an “oracle” setting with perfect retrieval. In this setting, we replay the ICL examples of the test task right before testing. We found that this is better than the original Lifelong ICL setting, but still lags behind the Single-task ICL baseline. This suggests that RAG can partly close the performance gap, but cannot fully solve the issue. See Table 3 and Figure 4 (Baseline vs. Recall vs. Replay) for more details.
>
> - Per requested, we experimented with a RAG baseline. In this baseline, we first segment context into task chunks. For a given test task, we use [an off-the-shelf retriever](https://huggingface.co/dunzhang/stella_en_1.5B_v5) to select the task chunk that best corresponds to the testing instruction. The relevant chunk is then prepended right before the test task. We conducted experiments on Mistral-7b and FILM-7b with 16 tasks and 4 shots. We report the results below.
> | Model | S-acc | L-acc | L-pass-rate | RAG-acc | RAG-pass-rate |
> |---|---|---|---|---|---|
> | Mistral-7b (32k) | 78.6 | 74.8  | 67.5 | 75.4 | 70.0 |
> | FILM-7b (32k) | 79.6 | 75.4  |  72.5 | 75.8  | 78.8  |
> - Consistent with your hypothesis, the RAG baseline achieves higher pass rate compared to the original Task Haystack pass rate. The accuracy of RAG is between that of Lifelong ICL and the "oracle" Replay setting, which is also expected.
>
>  > Additionally, could the authors conduct a similar controlled analysis for LLMs that are augmented by a naive RAG module? I think the community may care more about real long-context issues than those that could not be easily solved by RAG.
>
> - The goal of this paper is to stress-test *long-context* models and identify their limitations and undesired behaviors. While RAG methods provide reasonable references, they are beyond the scope of our work.
>
> - We agree with you that the long-context LM vs. RAG debate is a pressing and important one, and poses an open question in the community. We hope our analysis in Sec 5.2 and our new results with RAG models provide some clues for this question. We leave a controlled analysis for RAG methods as future work.
>
> We hope the above discussion addresses some of your concerns. Thank you once again for your thoughtful review and valuable feedback!

---

> > ### Comment · Reviewer_PEya · 2024-08-19
> >
> > Thanks for the detailed response.
> >
> > 1. Regarding the claim that the “Replay” setting be seen as an “oracle” setting with perfect retrieval, why do we need to keep "T1 Train T2 Train T3 Train" before "T1 Train T1 Test"? Is not the “oracle” setting with perfect retrieval just the Single-task ICL? For example, given the target task "T1 Test", we should just keep "T1 Train" in the context, right?
> >
> > 2. Based on the above point, may I ask how the RAG baseline is implemented in the additional results? Do you still keep "T1 Train T2 Train T3 Train" in the context similar to the "Replay" setting?
> >
> > 3. The additional results show that RAG outperforms Lifelong ICL, which does not resolve my concern about whether we need a long context here. I am still confused about how this benchmark is distinctly long context and qualitatively different from shorter context with RAG [1].
> >
> > [1] Is It Really Long Context if All You Need Is Retrieval? Towards Genuinely Difficult Long Context NLP

---

> > > ### Author Rebuttal · Authors · 2024-08-21
> > >
> > > Dear reviewer PEya,
> > >
> > > Thank you very much for getting back to us! We hope our response below can address your concerns.
> > >
> > > > Regarding the claim that the “Replay” setting be seen as an “oracle” setting with perfect retrieval, why do we need to keep "T1 Train T2 Train T3 Train" before "T1 Train T1 Test"? Is not the “oracle” setting with perfect retrieval just the Single-task ICL? For example, given the target task "T1 Test", we should just keep "T1 Train" in the context, right?
> > >
> > > > Based on the above point, may I ask how the RAG baseline is implemented in the additional results? Do you still keep "T1 Train T2 Train T3 Train" in the context similar to the "Replay" setting?
> > >
> > > We are sorry that we misinterpreted your comments in the earlier round. In the previous response, the RAG baseline was implemented as  “T1 Train, T2 Train, T3 Train, Retrieved Examples, T1 Test”. An alternative way to implement RAG, as you mentioned, is “Retrieved Examples, T1 Test”. We’ve added this experiment in the table below. We refer to the previous RAG as “Lifelong ICL + RAG”; and the newer RAG as “RAG”.
> > >
> > > | 16task-4shot acc | Single-task ICL (oracle) | RAG | Lifelong ICL | Lifelong ICL + RAG |
> > > |---|---|---|---|---|
> > > | Mistral-7b | 78.6 | 79.0  | 74.8 | 75.4 |
> > > | FILM-7b | 79.6 | 80.1 | 75.4 | 75.8  |
> > >
> > > (One minor detail is that for RAG we prepend the instruction to examples and test cases, so that the instruction appeared twice. In the Single-task ICL the instruction is only used once before the first example. This may explain the small gain from Single-task ICL to RAG)
> > >
> > > Consistent with your hypothesis, Task Haystack is a retrieval-style task that can be solved by RAG methods. However, we would like to argue that evaluating long-context models on retrieval-style tasks is still meaningful, and we discuss this further in the next paragraph.
> > >
> > > > The additional results show that RAG outperforms Lifelong ICL, which does not resolve my concern about whether we need a long context here. I am still confused about how this benchmark is distinctly long context and qualitatively different from shorter context with RAG [1].
> > >
> > > Thank you for raising your concern, and thanks for bringing up the highly relevant position paper [1], which calls for developing distinctly long-context benchmarks with high diffusion and large scope.
> > >
> > > We agree with you and [1] that developing distinctly long-context benchmarks is an important research direction. Meanwhile, we want to clarify that in our work **we start from a different question**: are long-context models behaving robustly and as expected? We do not claim Task Haystack to be “distinctly long-context”. It is indeed a retrieval-style task, but we believe **our work still makes meaningful progress** in long-context evaluation -- As discussed in page 5 of [1], “retrieval is still interesting”; as mentioned by reviewer 6Dct, one advantage of (retrieval-style) synthetic tasks is that it's much more controllable for ablations; as discussed in [2], long-context models have certain benefits over RAG methods, hence evaluating long-context models on retrieval-style tasks is still meaningful.
> > >
> > > [1] Is It Really Long Context if All You Need Is Retrieval? Towards Genuinely Difficult Long Context NLP. https://arxiv.org/abs/2407.00402
> > >
> > > [2] Can Long-Context Language Models Subsume Retrieval, RAG, SQL, and More? https://arxiv.org/abs/2406.13121
> > >
> > > **Additional thoughts on [1] - Do we need a third axis in the taxonomy?**
> > >
> > > The paper [1] introduces a taxonomy of long-context evaluation with two axes: (1) Diffusion: How hard is it to find and extract the necessary information?; (2) Scope: How long is the necessary information?
> > >
> > > Firstly, we really appreciate this work! It provides a new and systematic way of looking at long-context evaluation. However, when we attempt to put Task Haystack in this taxonomy, we find it challenging. Task Haystack is having low diffusion (it’s not hard to find relevant information) and small scope (the relevant information is short, only a few ICL examples), similar to the original NIAH, but is also more challenging than the original NIAH partly due to requiring contextual understanding (or “implicit aggregations” as briefly mentioned in [1]) of the context, as opposed to simple copying-and-pasting. Hence, we think it may be helpful to add a third axis of “contextual understanding / implicit aggregation” to the taxonomy, and Task Haystack can be seen as making progress in this third axis.
> > >
> > > We hope the above discussion addresses some of your concerns. Thank you once again for your valuable feedback!

---

> > > > ### Comment · Reviewer_PEya · 2024-08-22
> > > >
> > > > Thanks for your detailed reply including the additional RAG baseline and the discussions about the long-context evaluation. I am glad to see my hypothesis was verified.
> > > >
> > > > The debates surrounding the RAG versus long-context would benefit the field and your proposal “Do we need a third axis in the taxonomy?” sounds interesting.
> > > >
> > > > I appreciate all the efforts and I will raise my score to 6.

---

### Official Review · Reviewer_6Dct · 2024-07-23
**Interesting synthetic task but not yet convinced it provides new signal**

**Rating:** 7
**Confidence:** 4
**Correctness:** Seems plausible to me, except the poi…
**Clarity:** Yes, the writing is clear.

**Review:**

The work is high-quality, clear, and incrementally novel.

Pros: See strengths below

Cons: See opportunities for improvement

The main weakness from my perspective is that—given that this is a synthetic task—the only purpose of the task would be that it provides useful signal for developing models, beyond preexisting evals (NIAH and variants). I'm not really convinced that that's the case for the reasons described below.

Happy to increase the score to at least 6 if the weaknesses are addressed.

**Strengths:**

1. Lifelong ICL task is interesting in that it hybridizes retrieval and ICL for NIAH.

2. Mostly nice evaluation protocol (sampling 5 shots, comparing performance relative to short-context for memorization, etc.)

3. Some interesting ablations (e.g. "multi-epoch ICL").

4. Honest discussion of limitations

**Additional Feedback:**

On line 121, I think it would be more correct as $\hat{y} \sim LM(...)$ rather than $\hat{y} = LM(...)$ since $\hat{y}$ is a sample rather than the entire distribution.

When you write "s-acc" and “l-acc", it would be clearer as capital letters S-acc and L-acc. Lowercase L looks like a #1.

**Documentation:**

Yes

**Limitations:**

Yes, limitations as a synthetic task, one language, unimodal, etc. are described.

**Opportunities For Improvement:**

1. I don't think the task really delivers on the promise of "real-world applications of long-context models, such as a personal assistant, where models encounter shifting topics and may need to resume from earlier threads in the context", since the tasks are just concatenated in chunks which isn't how users interact with models. It seems like a natural generalization to interleave examples across tasks too, and you could control the degree to which this happens to set the difficulty, like multiple NIAH. As it is, it still has the synthetic NIAH flavor where the model is picking out one section of the context window (a single task) and ignoring all the rest.

2. I am a little skeptical of how much new signal this provides compared to NIAH. First, it would be nice to compare against the concrete NIAH numbers reported for each of these models, or compute your own. (Hopefully I didn't just miss these numbers somewhere in the paper.) I'm not sure that they actually report "near-perfect" numbers, except Gemini 1.5 and maybe GPT-4o. Second, the gap between S-acc and L-acc, while apparently statistically significant, seems pretty small in absolute. And there are some weirdnesses like Llama 2-7B (80k) and GPT-4o having higher L-acc than S-acc, but non-100% pass rate. If you flipped L and S, S would probably have a non-100% pass rate too. This seems like a problem with the metric being asymmetric (pass rate rather than win rate), or maybe your significance threshold for the t test doesn't take into account that you are performing many t tests in parallel.

3. I won't hold it against you since I think Gemini 1.5 became officially available near the submission deadline, but experiments on it would improve the paper given that it is the best model for long context by a wide margin.

4. I think this particular data is unlikely to be used as a benchmark because many of the sub-datasets have unspecified licenses. You might want to provide a gold subset of datasets with clear and permissive licenses that could be the standard.

**Relation To Prior Work:**

Related work is discussed but could be a bit better.

For realistic long context evaluations, you could mention MTOB (https://arxiv.org/abs/2309.16575), Infinity-Bench (https://arxiv.org/abs/2402.13718), SWE-Bench (https://arxiv.org/abs/2310.06770), etc. There are probably more "agent" benchmarks like SWE-Bench that test long contexts in effect, even if they're not framed primarily in terms of long context. You could also mention more realistic multimodal long context tasks (e.g. audio, video QA, long ASR) which tend to use more tokens. You might want to mention perplexity curves (x axis = sequence length) being used to evaluate long context LLMs.

For the synthetic ones, there are some citations throughout the paper that should probably be mentioned in the synthetic benchmark section (Liu et al. 2024b, Levy et al. 2024).

I think it would also make your point "struggles to scale, especially for tasks requiring million-token contexts" more concrete if you mentioned the number of tokens for each of the realistic tasks (and then the synthetic tasks are unlimited length). I don't think this really applies within the regime that you're working with (<32k tokens). The real advantage of synthetic tasks here is that it's much more controllable for ablations.

**Summary And Contributions:**

The main contribution is a version of NIAH that is lifted up a level, to retrieve one of many tasks (instructions+ICL examples) in context and perform that task. This task is evaluated across a number of models and ablated to give some insight into the source of the performance regression (recency, distraction, context length, zero-shot abilities, etc.).

---

> ### Author Rebuttal · Authors · 2024-08-17
>
> Dear reviewer 6Dct,
>
> Thank you for your insightful review! You’ve raised some important discussion points and we’re truly grateful for your help in making our paper better.
>
> > How much new signal this work provides compared to NIAH and variants.
>
> - **Task Haystack clearly provides new signals compared to the naive NIAH**. Please refer to the result table in the “Reporting NIAH results” section below.
>
> - **The focus of Task Haystack is distinct from other NIAH variants**. The core idea of Task Haystack is to “level up” the NIAH to the task level and test the models’ contextual utilization (i.e., in-context learning) beyond simple copying-and-pasting. This is orthogonal to NIAH variants such as multi-needle or multi-hop variable tracing, as done in the RULER benchmark (https://arxiv.org/abs/2404.06654).
>
> - **We found cases where Task Haystack provides new signals compared to RULER, a benchmark of NIAH variants**. For example, RULER suggests that Command-R-35B and Yi-34B have an effective context length of 32k, but these two models achieve unsatisfactory pass rates of 41.2% and 53.8% in our 32k experiments.
>
> - It’s possible that the performance on different long-context abilities/benchmarks are correlated, but this requires further (meta-)investigation and is, in itself, an interesting research question. To support such studies, we will continue evaluating more models, including future ones, on Task Haystack.
>
> > Whether the benchmark delivers the promise of “real-world applications of long-context models, …, where models encounter shifting topics and may need to resume from earlier threads in the context”.
>
> Our main argument is that previous benchmarks do not adequately capture the aspect of “shifting topics in the context,” which may be a challenge in real-world applications of long-context models. This makes our benchmark a reasonable proxy for such applications. We will rephrase this part to avoid the confusion.
>
> > “It seems like a natural generalization to interleave examples across tasks too …”
>
> Thanks for the suggestion! It’s a good idea to interleave examples to make it similar to a multi-needle haystack and make the benchmark more difficult. However, the fact that many models struggle on our current single-needle design is concerning, so we decide to limit the scope to this easier setting, and conduct further analysis and ablation in this setting.
>
> It will be definitely interesting to study the interleaved variation, as well as other variations to control the difficulty. In the following, we conduct preliminary experiments of interleaving examples (termed as Multi-task ICL and M-acc), and find models to perform worse than Lifelong ICL in this setting. We leave further investigation as future work.
>
> | 16-Task, 4-shot | S-acc | L-acc | M-acc |
> |----------------------|------|-----|-----|
> | Mistral-7B (32k) | 78.6  | 74.8 | 72.1 |
> | FILM-7B (32k) | 79.6  | 75.4 | 74.7 |
>
> > Reporting NIAH results
>
> We’ve experimented with the basic version of NIAH (https://github.com/gkamradt/LLMTest_NeedleInAHaystack), and visualized the results in Figure 9-16 in the submission. In the table below we report the pass rates in numbers, and we will include them into the paper.
>
> | Model            | Pass (%) | Model            | Pass (%) |
> |------------------|----------|------------------|----------|
> | Mistral-7B (32k) | 95.3  | Llama3-8b (1048k) | 100.0 |
> | FILM-7B (32k) | 100.0 | Yi-6B (200k) | 100.0 |
> | Llama2-7B (32k) | 95.3  | Yi-9B (200k) | 100.0 |
> |  Llama2-7b (80k) | 100.0  | Yi-34B (200k) | 100.0  |
>
>
> We would like to mention that several compared models claim near-perfect NIAH results and use these as their selling point (e.g., https://arxiv.org/abs/2402.10171 Figure 1; https://huggingface.co/gradientai/Llama-3-8B-Instruct-Gradient-1048k; https://arxiv.org/pdf/2403.04652 Figure 6; etc.)
>
> Regarding harder NIAH variants, we believe the focus of Task Haystack is distinct from them and thus provides additional value.
>
> > Concerns of the asymmetric design of the pass rate
>
> - To clarify, it is possible that L-acc is higher than S-acc but the pass rate is low. In our definition, in a 16-task haystack with 80 cases (5 permutations x 16 tasks), pass rate measures how often L-acc is not significantly worse than S-acc among the 80 cases. We anticipate the model to pass in *all* 80 cases (i.e., 100% pass rate). Meanwhile, L-acc minus S-acc is capturing the *average* of the 80 cases.
>
> - We also want to clarify that the pass rate is *not* analogous to the commonly-used win rate. In the terminology of win rate, both win (L-acc significantly better than S-acc) and tie (no significant differences) are considered as acceptable/“passing”, but lose is not acceptable.
>
> - The metric was not designed for symmetry. While it is possible to flip S-acc and L-acc and compute pass rates, this would not be the intended use. We will explain this further in the paper to avoid any confusion.
>
> > Experiments on Gemini 1.5
>
> Thank you for the suggestion! We are working on evaluating Gemini 1.5 and will report the results here once we have them.
>
> During the reviewing period we have evaluated several other models: Llama-3-70B (1048k), Command-R-35B (128k) and Llama-3.1-70B (128k). We report a snippet of the results (16-task 8-shot) below, and will include the full results in the future version of the paper.
>
> |                      | S-acc | L-acc | Pass |
> |----------------------|-------|-------|------|
> | Llama3-70B (1048k)   | 81.7 | 75.7 | 51.2 |
> | Command-R-35B (128k) | 80.5 | 75.3 | 41.2 |
> | Llama3.1-70B (128k)  |  85.2  | 83.3  |  80.0  |

---

> > ### Author Rebuttal · Authors · 2024-08-17
> >
> > > Dataset Licensing
> >
> > Thank you for the suggestion! Proper data use is a priority for us, but it can be challenging due to insufficient documentation in some past dataset releases. For example, SST-2 is a widely used dataset, yet we have been unable to find any licensing information for it.
> >
> > Based on your suggestion, we plan to do the following:
> >
> > (1) We will create a subset of 16 tasks that has permissive licenses, and encourage future users to use this subset. We will also rerun selected models on this subset. This will take some time and we will update the paper once we have the results.
> >
> > (2) We will explicitly mention in the paper and in our codebase that we allow dataset authors to opt out.
> >
> > > Suggestions on related works
> >
> > Thank you for suggesting these highly-relevant works, particularly the agent benchmarks like SWE-bench, which also require long-context understanding capabilities, but in a less obvious way. Also, thank you for suggesting to reference the input lengths of existing realistic benchmarks and highlight the controllability of synthetic tasks. We will update our related work section to include these references and provide a more detailed discussion in the appendix.
> >
> > Thank you once again! We truly appreciate your insights. Please don’t hesitate to reach out if you have any follow-up questions.

---

> > > ### Comment · Reviewer_6Dct · 2024-08-19
> > >
> > > Thanks for the detailed response, I've increased my review score to 7 due to the NIAH baselines, promise to include Gemini 1.5, clearly-licensed gold subset, and related work updates. I think it is reasonable to leave detailed study of interleaving to future work.
> > >
> > > I am still a bit confused about the pass rate definition. If you flipped S-acc and L-acc, would you currently get a pass rate of 100%? (i.e., "does short context do as well as long context?") Do the thresholds for the t tests incorporate the number of tests that are being performed in parallel? I don't see p values mentioned anywhere.

---

> > > > ### Author Rebuttal · Authors · 2024-08-21
> > > >
> > > > Dear reviewer 6Dct,
> > > >
> > > > Thank you very much for getting back to us! Regarding your follow-up questions:
> > > >
> > > > > If you flipped S-acc and L-acc, would you currently get a pass rate of 100%? (i.e., "does short context do as well as long context?")
> > > >
> > > > No. If we flip S-acc and L-acc, the pass rate will not be 100%.
> > > >
> > > > To better clarify, we present a concrete example with 16 tasks. As we use 5 different permutations of task orders, there are 80 cases in total. In each case, we conduct a two-sided t-test (over 5 different samples of ICL examples), which results in either (1) Win: L-acc significantly better than S-acc; (2) Tie: no significant difference; (3) Lose: L-acc significantly worse than S-acc. Consider that there are 10 cases in (1), 40 cases in (2) and 30 cases in (3). In the normal setting, the pass rate = (10+40)/80 = ⅝. In the flipped setting, the pass rate = (40+30)/80=⅞.
> > > >
> > > > Firstly, we would like to clarify that it’s not recommended to compute pass rates in the flipped setting. Secondly, for the remaining ⅛ cases, L-acc is significantly better than S-acc, which is possible and can be seen as the model benefiting from positive task transfer. See our section 5.3 “Trends of positive forward and backward transfer” for more discussion on this.
> > > >
> > > > > Do the thresholds for the t tests incorporate the number of tests that are being performed in parallel? I don't see p values mentioned anywhere.
> > > >
> > > > Thank you for your very careful review! Currently we consider p<0.05 as significant for each t-test, but we now learned that this design doesn't account for the effect of multiple t-tests, which could lead to an increased risk of Type I errors. We recognize this as a limitation and conduct more investigation into this.
> > > >
> > > > We considered applying Bonferroni Correction, a common approach that accounts for the effect of multiple tests, by adopting 0.05/80=0.000625 as the critical p-value. However, this introduces new challenges, as this naive fix increases the risk of Type II errors, and can lead to overestimated pass rates. The table below shows the pass rates of selected models before and after applying the correction.
> > > >
> > > > | 16-task 4-shot Pass Rate | Before | After |
> > > > |---|---|---|
> > > > | Mistral-7B (32k) | 67.5 | 92.5 |
> > > > | FILM-7B (32k) | 72.5 | 96.3 |
> > > > | Llama3-7B (1048k) | 71.3 | 97.5 |
> > > > | Yi-6B (200k) | 43.8 | 83.8 |
> > > > | GPT-4o (128k) | 83.8 | 100.0 |
> > > >
> > > > We also considered Benjamini-Hochberg Correction which creates a different trade-off between Type I and Type II errors. Our concern is that this method leads to different p-value thresholds for different models, which complicates the evaluation and may lead to unfair comparison across models.
> > > >
> > > > At this point, we believe there’s no perfect correction to be applied here. One way of interpreting the p-value threshold is the trade-off it creates between Type I and Type II errors, and thus it controls the calibration of the pass rates we report. Hence, the impact of using p<0.05, rather than a lower threshold, is that the pass rates are calibrated to be lower and more conservative across all experiments. It affects the quantities, but the qualitative conclusions in the paper are not expected to change. We plan to maintain the current design but will ensure that these limitations are thoroughly discussed.
> > > >
> > > > We hope the above discussion addresses some of your concerns. Thank you once again for your valuable feedback!

---

> > > > > ### Author Rebuttal · Authors · 2024-08-31
> > > > >
> > > > > Dear reviewer 6Dct,
> > > > >
> > > > > Following your suggestions, we have recently conducted experiments (1) using Gemini 1.5 Flash; (2) using a subset of tasks with permissive licenses. Below is a brief summary of the results.
> > > > >
> > > > > > Experiments on Gemini 1.5
> > > > >
> > > > > We have conducted experiments with Gemini 1.5 Flash, in the Scale-Shot setting (16 tasks, 1/2/4/8-shot). Due to budget constraints, we have not been able to run experiments on the Gemini 1.5 Pro model yet. Gemini 1.5 Flash achieves competitive performance, but it lags slightly behind GPT-4o.
> > > > >
> > > > > | 16-task | 1-shot |       |           | 2-shot |       |           | 4-shot |       |           | 8-shot |       |           |
> > > > > | :-----: | ------ | ----- | --------- | :----: | ----- | --------- | ------ | ----- | --------- | ------ | ----- | --------- |
> > > > > |         | S-acc  | L-acc | pass-rate | S-acc  | L-acc | pass-rate | S-acc  | L-acc | pass-rate | S-acc  | L-acc | pass-rate |
> > > > > | Gemini 1.5 Flash | 78.0  | 79.1 | 87.5     | 77.9  | 79.4 | 87.5     | 79.4  | 80.4 | 85.0     | 77.9  | 81.6 | 80.0     |
> > > > >
> > > > > > Tasks with Permissive Licenses
> > > > >
> > > > > We have sampled a subset of 16 tasks, each with a permissive license. The table below outlines these tasks and their respective licenses. In the future, we will encourage users of Task Haystack to use this subset for benchmarking and comparison.
> > > > >
> > > > > |Tasks | License| Tasks | License |
> > > > > |-|-|-|-|
> > > > > |metaphor-boolean, amazon-massive-scenario, acl-arc | Apache 2.0| fever, dbpedia14 | CC BY-SA 3.0|
> > > > > |function-of-decision-section, brag-action, semeval-absa-laptop| CC BY 4.0|silicone-dyda-da | CC BY-SA 4.0|
> > > > > |climate-commitments-actions, environmental-claims | CC BY-NC-SA 4.0|wic | CC BY-NC 4.0|
> > > > > |student-question-categories|CC0|wiki-hades|MIT|
> > > > > |senteval-cr, babi-nli| BSD|
> > > > >
> > > > > We have rerun selected models on this subset.
> > > > >
> > > > > | Model | 1-shot |       |           | 2-shot |       |           | 4-shot |       |           | 8-shot |       |           |
> > > > > | :-----: | ------ | ----- | --------- | :----: | ----- | --------- | ------ | ----- | --------- | ------ | ----- | --------- |
> > > > > |         | S-acc  | L-acc | pass-rate | S-acc  | L-acc | pass-rate | S-acc  | L-acc | pass-rate | S-acc  | L-acc | pass-rate |
> > > > > |Mistral-7b| 67.5| 68.9| 95.0|70.7|69.4|70.0|72.3|69.7|52.5|-|-|-|
> > > > > |FILM-7b|68.9|71.1|91.2|71.4|71.9|87.5|72.7|72.4|73.8|-|-|-|
> > > > > |Llama3.1-70b|73.8|74.2|83.8|75.0|75.1|88.7|76.6|75.7|77.5|77.5|75.6|73.8|
> > > > >
> > > > >
> > > > > (Note: “-” indicates that the prompt exceeds the maximum context length.)
> > > > >
> > > > > ---
> > > > >
> > > > > We will incorporate the results above into our paper. Thank you once again for your valuable feedback! We enjoyed our discussion with you.

---

### Official Review · Reviewer_mKcD · 2024-07-24
**review for Stress-Testing Long-Context Language Models with Lifelong ICL and Task Haystack**

**Rating:** 7
**Confidence:** 3
**Correctness:** The paper is mostly correct.
**Clarity:** The paper is mostly clear.

**Review:**

This paper overall has clear contribution, and is addressing a very relevant question at hand. The Task Haystack suit is innovative, which is designed to evaluate long-context LMs on their ability to utilize long contexts effectively, avoiding distractions, and achieving test accuracy comparable to single-task ICL baselines. The model list is comprehensive which incorporates the SOTA LLMs at the moments.

**Strengths:**

The evaluation is poised at solving the issues of 'copy and paste' from context as a retrieval task but measures the model capacity to reason within the given examples. The paper benchmarks a variety of state-of-the-art long-context LMs, including both closed and open models. This extensive benchmarking provides valuable insights into the current capabilities and limitations of these models.

**Additional Feedback:**

n/a

**Documentation:**

The documentation is clear. But it's not clear where is the shared code and data

**Ethics:**

There's no major ethic concern.

**Limitations:**

While the evaluation protocol accounts for instabilities in ICL experiments, the paper could explore more robust statistical methods or a larger number of permutations to strengthen the reliability of their findings.

**Opportunities For Improvement:**

While Task Haystack provides a robust framework for evaluation, the paper does not address potential scalability issues or how this suite can be applied to tasks beyond classification. Including diverse types of tasks and more complex real-world scenarios could enhance the evaluation’s comprehensiveness.

**Relation To Prior Work:**

Discussion of prior work is sufficient. The paper addresses the gap of prior work and contribution of this paper.

**Summary And Contributions:**

This paper introduces a life long in context learning evaluation pipeline for LLMs, which is a known challenge given the phenomenon of catastrophic forgetting. The pipeline involves learning many different tasks, and evaluating on many different metrics, to showcase the current LLMs still fall short of giving reasonable results on lifelong ICL.

---

> ### Author Rebuttal · Authors · 2024-08-17
>
> Dear Reviewer mKcD,
>
> Thank you for your insightful review! We appreciate that you find our paper clear, relevant and comprehensive. We’re grateful for your helpful comments and have responded to each below.
>
> **Potential Scalability Issues**
> >While Task Haystack provides a robust framework for evaluation, the paper does not address potential scalability issues
>
> Our framework is designed to scale flexibly by incorporating more shots and more tasks. Please refer to Sec 4 “Controlling the Context Length” for more details. For example, the context can extend to 128k tokens with 64 tasks and 8 shots. And it’s possible to include more tasks into the current collection. Could you please explain a bit more on your concern of scalability?
>
> **Including Task Types Beyond Classification**
> >..or how this suite can be applied to tasks beyond classification
>
> Thank you for your suggestion. As described in Section 3, our framework admits any language task that includes an instruction and input-output pairs. The framework can easily integrate other task types, such as summarization, with their respective evaluation metrics, like ROUGE or BLEU scores. However, to operationalize task selection on our side, and to simplify and standardize the evaluation process, we opt to use classification tasks in the current version of Task Haystack.
>
> **Including more complex real-world scenarios**
>
> We understand your concern. We think developing long-context evaluation suites reflecting real-world needs is critical yet challenging. Currently there is limited understanding of how human users interact with long-context LLMs in real-world settings.
>
> In the future, we hope that a study similar to WildChat (https://arxiv.org/pdf/2405.01470) can be conducted with a focus on long-context models. Such research could expand our understanding of the real-world usage of these models and enable more realistic evaluations similar to WildBench (https://github.com/allenai/WildBench).
>
> **Robust Statistical Methods and More Permutations**
>
> Thanks for your suggestion! If you could further elaborate on your ideas for developing more robust statistical methods, we would be happy to discuss them and consider incorporating them into our benchmark.
>
> We agree with you that increasing the number of permutations can reduce the randomness of permutation sampling. However, this also increases computational cost. Our goal is to ensure that our benchmark is both reliable and cost-effective. To explore this balance, we conducted additional experiments with a higher number of permutations (10 and 20) on selected models.
>
> |     Permutations       | 5 | | 10 | | 20 | |
> |:----------:|:-------:|:------:|:--------:|:-------:|:--------:|:-------:|
> | **Model**  | **L-acc** | **pass-rate** | **L-acc** | **pass-rate** | **L-acc** | **pass-rate** |
> | Mistral-7B (32k) | 74.2    | 47.5 | 74.6     | 49.4  | 74.8     | 50.6  |
> | FILM-7B (32k)    | 74.9    | 55.0 | 75.1     | 55.6  | 75.2     | 54.1  |
> | Llama2-7B (80k)  | 61.5    | 76.3 | 61.3     | 78.7  | 60.5     | 77.5  |
> | Llama3-8B (1048k)| 70.1    | 57.5 | 69.9     | 56.9  | 70.0     | 56.6  |
>
> The differences between 5 and more permutations are minimal (<3%). Furthermore, the ranking of models remains consistent across these settings. Hence we believe using 5 permutations provides a reasonable balance between reliability and computational efficiency.
>
> **Shared Code and Data**
>
> We have uploaded our code and data in the supplementary material. If there’s any issue with access, please contact us and the area chair again! We will open-source our code and data upon acceptance of paper.
>
> We hope the above discussion addresses some of your concerns. Thank you once again for your thoughtful review and valuable feedback!

---

### Official Review · Reviewer_seb7 · 2024-07-26
**A benchmark to evaluate the lifelong in-context learning abilities of long context LMs**

**Rating:** 6
**Confidence:** 4
**Correctness:** Yes, it's correct.
**Clarity:** Yes, the paper is well written.

**Review:**

The paper is clear. The topic of the paper is important.

**Strengths:**

1. The paper is well-written and easy to follow.
2. The topic of lifelong in-context learning is interesting and important.
3. The experiments are comprehensive and clear.

**Additional Feedback:**

Please check Opportunities For Improvement above.

**Documentation:**

Yes.

**Limitations:**

Yes, the limitations are included.

**Opportunities For Improvement:**

1. It's interesting to test the model's performance on multiple tasks instead of only one task in the task haystack. The ability to retrieve multiple tasks is also important and interesting.
2. The tasks may not necessarily be fewer than 20 categories. With language models' context lengths getting longer and longer, it's also interesting to test whether the model can select the correct task from the haystack and effectively leverage the long demonstration within the task.

**Relation To Prior Work:**

Yes.

**Summary And Contributions:**

This paper proposes Lifelong ICL, a benchmark to evaluate the lifelong in-context learning abilities of long-context LLMs. The benchmark  leverages 64 existing datasets to create the linelong task haystack toevaluate the model using long in-context inputs. The paper also analyses existing long-context LLMs and give detailed comparisons.

---

> ### Author Rebuttal · Authors · 2024-08-17
>
> Dear Reviewer seb7,
>
> Thank you for your insightful review! We appreciate your recognition of the paper’s clarity and the importance of the topic discussed. We’re grateful for your helpful comments and have responded to each below.
>
> **Testing on multiple tasks**
> > It's interesting to test the model's performance on multiple tasks instead of only one task in the task haystack. The ability to retrieve multiple tasks is also important and interesting.
>
> Thank you for your suggestion! To clarify, our current benchmark already tests each model on multiple tasks *individually* at inference time. If you refer to simultaneous inference on multiple examples (from different tasks), in a way similar to batched inference (https://arxiv.org/abs/2301.08721) or multi-task inference (https://arxiv.org/abs/2402.11597), we believe these are valid paths to further increase the complexity of Task Haystack. However, given that the current Task Haystack already presents significant challenges to most models, we plan to leave this as future work.
>
> **Incorporating Classification Tasks with More than 20 Categories**
> >The tasks may not necessarily be fewer than 20 categories. With language models' context lengths getting longer and longer, it's also interesting to test whether the model can select the correct task from the haystack and effectively leverage the long demonstration within the task.
>
> Thank you for raising this concern! We introduce this selection criteria mainly due to a finding in prior work (https://arxiv.org/abs/2404.02060), where tasks with an excessive number of categories can pose great challenges to most models on their own. Incorporating one task with an excessively long prompt could potentially overshadow the primary objectives of our benchmark. Therefore, we consciously chose to limit the number of categories.
>
>
> We anticipate that Task Haystack will be a living benchmark that evolves with the progress of long-context LMs development. Your suggestions on increasing the complexity of Task Haystack are valuable for our future research. That being said, we think it is equally important to benchmark and investigate long-context LMs in the current Task Haystack as they still face significant challenges here.
>
> Thank you once again for your thoughtful review and valuable feedback!

---

### Author Rebuttal · Authors · 2024-08-17

Dear reviewers,

Thank you for your valuable feedback! We sincerely appreciate your efforts in helping us improve our paper! Additionally, we are encouraged that you find our benchmark to be interesting (Reviewer 6Dct, seb7) and timely (PEya, mKcD), our analysis to be comprehensive and rigorous (PEya, seb7), and our paper to be clear and well-written (6Dct, mKcD).

In your reviews, we identified two primary concerns:

(1) **Whether Lifelong ICL and Task Haystack is a realistic problem setting (6Dct and PEya)**. We would like to clarify that we are proposing a *hybrid* benchmark focused on stress-testing long-context LMs. While it incorporates realistic elements, we do not claim it to be a fully realistic benchmark at this stage. We believe this is a confusion arising from our imprecise phrasing and we will revise the paper to better reflect our intentions. Please refer to our responses to 6Dct and PEya for more details.

(2) **Whether Lifelong ICL and Task Haystack provide new signals to long-context LM evaluation (6Dct)**. Empirically, we show that Task Haystack provides new information compared to the naive NIAH and its variants. Additionally, we emphasize that Task Haystack targets specific aspects not addressed by previous benchmarks. Please refer to our response to 6Dct for more details.


Over the past week, we have conducted additional experiments to address the questions raised.

(1) **Number of task permutations (seb7)**: We show that using 5 task permutations leads to results consistent with those using 10 or 20 permutations, while being more cost-effective.

(2) **RAG-baseline (PEya)**: We have implemented a RAG method that uses retrieved examples after the Lifelong ICL prompt and before the test cases.

(3) **Interleaving examples of different tasks (6Dct)**: We conduct preliminary experiments in a more challenging setting where ICL examples of different tasks are interleaved in the context.

Please refer to our responses to individual reviewers for more detailed information. We would be glad to discuss any follow-up questions or additional comments you may have. Thank you once again!

Authors of Submission 2319

---

### Decision · Program_Chairs · 2024-09-26

**Decision:**

Accept (Poster)

**Comment:**

This paper proposes a benchmark to study how the long-context LLM use long contexts and  lifelong ICL. All reviewers provide positive score for this paper and recognize the novelty and importance of this benchmark. AC read rebuttal and reviews and agreed with their decision.